# Consistent responses of vegetation gas exchange to elevated atmospheric $CO_2$ emerge from heuristic and optimization models

Stefano Manzoni[1,2], Simone Fatichi[3], Xue Feng[4,5], Gabriel G. Katul[6,7], Danielle Way[7,8,9], Giulia Vico[10]

[1]Department of Physical Geography, Stockholm University, Stockholm, SE-106 91, Sweden
[2]Bolin Centre for Climate Research, Stockholm University, Stockholm, SE-106 91, Sweden
[3]Department of Civil and Environmental Engineering, National University of Singapore, Singapore
[4]Department of Civil, Environmental, and Geo-Engineering, University of Minnesota, Minneapolis, MN 55455, USA;
[5]Saint Anthony Fall Laboratory, University of Minnesota, Minneapolis, MN 55455, USA
[6]Department of Civil and Environmental Engineering, Duke University, Durham, NC, 27708-0287, USA
[7]Nicholas School of the Environment, Duke University, Durham, NC, 27708 USA
[8]Department of Biology, University of Western Ontario, London, Ontario, N6A 5B7, Canada
[9]Environmental & Climate Sciences Department, Brookhaven National Laboratory, Upton, NY, 11973 USA
[10]Department of Crop Production Ecology, Swedish University of Agricultural Sciences (SLU), Uppsala, SE-750 07, Sweden

*Correspondence to*: Stefano Manzoni (stefano.manzoni@natgeo.su.se)

**Abstract.** Elevated atmospheric $CO_2$ concentration is expected to increase leaf $CO_2$ assimilation rates, thus promoting plant growth and increasing leaf area. It also decreases stomatal conductance, allowing water savings, which have been hypothesized to drive large-scale greening, in particular in arid and semiarid climates. However, the increase in leaf area could reduce the benefits of elevated $CO_2$ concentration on soil water depletion. The net effect of elevated $CO_2$ on leaf- and canopy-level gas exchange remains uncertain. To address this question, we compare the outcomes of a heuristic model based on the Partitioning of Equilibrium Transpiration and Assimilation (PETA) hypothesis and a model based on stomatal optimization theory. Predicted relative changes in leaf- and canopy-level gas exchange rates are used as a metric of plant responses to changes in atmospheric $CO_2$ concentration. Both models predict reductions of leaf-level transpiration rate due to decreased stomatal conductance under elevated $CO_2$, but negligible (PETA) or no (optimization) changes in canopy-level transpiration due to the compensatory effect of increased leaf area. Leaf- and canopy-level $CO_2$ assimilation are predicted to increase, with an amplification of the $CO_2$ fertilization effect at the canopy-level due to the enhanced leaf area. The expected increase in vapor pressure deficit (VPD) under warmer conditions is predicted to decrease the sensitivity of gas exchange to atmospheric $CO_2$ concentration in both models except at growth temperatures lower than the photosynthetic thermal optimum. The consistent predictions by different models that canopy-level transpiration varies little under elevated $CO_2$ due to combined stomatal conductance reduction and leaf area increase highlights the coordination of physiological and morphological characteristics in vegetation to maximize resource use (here water) under altered climatic conditions.

# 1    Introduction

Elevated atmospheric $CO_2$ causes stomatal closure and reduces transpiration while increasing net $CO_2$ assimilation at the leaf-level (Medlyn et al., 2001). These leaf-level observations led to the hypothesis that whole plant-, stand-, or catchment-scale transpiration would also be reduced as a consequence of increasing atmospheric $CO_2$ concentrations. Results from Earth system models (Fowler et al., 2019; Mankin et al., 2019; Betts et al., 2007; Swann et al., 2016) seem to support this hypothesis. Nevertheless, empirical evidence of decreased transpiration based on runoff measurements is limited (Ukkola et al., 2016). This discrepancy may be explained by the fact that Earth system models do not always include all the indirect effects of elevated $CO_2$ on plants (De Kauwe et al., 2021), such as increased plant growth and thus leaf area (Pan et al., 2022; Norby et al., 1999). Higher growth is also in part stimulated indirectly via reduced transpiration rate and hence less frequent water stress. Leaf area has been observed to increase the most in water-limited ecosystems (Donohue et al., 2013) and in open canopies (Bader et al., 2013; Duursma et al., 2016), but it increased also in some mesic forests (McCarthy et al., 2006; Norby et al., 1999), as well as in crops and herbaceous natural vegetation (Pritchard et al., 1999). This increase in the canopy-level evaporating surface area could counterbalance the reduction in leaf-level transpiration caused by stomatal closure, but it is not clear if and under which conditions these two effects balance out.

There is empirical evidence for the compensatory effects of stomatal closure and leaf area increase on canopy-level transpiration under elevated $CO_2$. The compensatory effect has been observed in water-limited ecosystems, where total evapotranspiration is already at its upper limit (Donohue et al., 2013; Schymanski et al., 2015), as well as in mesic forests, where transpiration rates can be insensitive to atmospheric $CO_2$ (Tor-ngern et al., 2015; Schäfer et al., 2002). More generally, canopy transpiration rates are unaffected or can even increase under elevated atmospheric $CO_2$ when the canopy is relatively open (leaf area index, LAI<5 $m^2$ $m^{-2}$, Donohue et al. (2017)). Similarly at the catchment scale, evapotranspiration did not change significantly with increasing $CO_2$ concentrations, as evidenced by minor variations in runoff attributed to trends in atmospheric $CO_2$ (Knauer et al., 2017; Yang et al., 2021). All these findings suggest that the net effect of increasing atmospheric $CO_2$ concentration on canopy transpiration appears lower than its effect at the leaf level.

In line with these empirical results, a detailed process-based model predicted that the direct effect of elevated atmospheric $CO_2$ on stomatal conductance is likely to be compensated by the indirect effects of higher evaporative demand through larger leaf area, especially in dry and semi-arid regions (Fatichi et al., 2016, 2021). In particular, elevated atmospheric $CO_2$ did not affect evapotranspiration at dry sites, and caused a small decline (-4 to -7%) at wet or intermediately wet sites, where increases in leaf area did not significantly improve light capture (Fatichi et al., 2016). Similarly, an optimality-based model showed that reduced stomatal conductance in response to elevated $CO_2$ was offset by increased leaf area mainly in water-limited environments with low canopy coverage, whereas such a compensatory effect did not emerge in energy-limited environments (Schymanski et al., 2015). When considering plant acclimation to elevated $CO_2$ using the same model, evapotranspiration in water-limited ecosystems even increased because of deepening roots and reduced bare soil evaporation due to shading. Finally, only partial compensation by leaf area was predicted by the model DESPOT, resulting in lowering of

canopy-level transpiration under elevated $CO_2$ (Buckley, 2008). Therefore, empirical and modelling results consistently point to some compensation of leaf-level stomatal downregulation by increased leaf area, at least in water limited systems and in young stands. Nevertheless, the question remains as to how the net effect of elevated atmospheric $CO_2$ on canopy-level gas exchange varies across ecosystems when $CO_2$ concentrations change in concert with increasing vapor pressure deficit (VPD, or $D$) and soil aridity.

Compared to complex process-based models, parsimonious analytical models can provide more direct understanding and theoretical insight into this question. Analytical models of plant gas exchange have been formulated based on different assumptions, ranging from heuristic relationships to eco-evolutionary theory. An example of the first type is the heuristic Partitioning of Equilibrium Transpiration and Assimilation (PETA) model, which describes how leaf area index (LAI), canopy and leaf transpiration, and $CO_2$ assimilation are expected to vary in response to elevated atmospheric $CO_2$

concentrations (Donohue et al., 2017, 2013). This model is based on the observation that leaf-level water use efficiency increases linearly with atmospheric $CO_2$ concentration, and assumes a set of relations between the relative changes in $CO_2$ assimilation and transpiration rates, as well as between the relative changes in climatic conditions (e.g., VPD) and leaf area associated with increasing atmospheric $CO_2$ concentrations. An alternative approach is to consider plant responses to changes in environmental conditions as optimized by natural selection (Harrison et al., 2021). Along these lines, optimal

stomatal conductance models were developed on the assumption that net $CO_2$ assimilation is maximized due to stomatal regulation of gas exchange (Cowan and Farquhar, 1977; Mencuccini et al., 2019). Both heuristic and optimization approaches provide closed-form solutions for gas exchange rates as a function of environmental conditions and plant characteristics, illustrating in a transparent way the compound effects of atmospheric $CO_2$ concentrations and other climatic conditions such as VPD and soil aridity. However, predictions from these two analytical models have never been compared.

Optimal stomatal conductance models are sensitive to changes in atmospheric $CO_2$ to different degrees depending on how they are formulated. Among the numerous models available (Mencuccini et al., 2019; Wang et al., 2020, and references therein), we focus here on those formulated as an optimal control problem in which stomatal conductance is solved through time. In these models, $CO_2$ responses depend on how the net $CO_2$ assimilation rate is represented and how the Lagrange multiplier for the optimization problem ($\lambda$, interpreted as marginal water use efficiency) is set (Katul et al., 2010; Medlyn et

al., 2011; Buckley and Schymanski, 2014). A key limitation of these optimization approaches is that $\lambda$ remained unspecified and has thus been regarded as a fitting parameter, because changes in soil water availability during dry periods have not been explicitly considered. This approach is equivalent to performing an 'instantaneous' optimization without considering the soil water dynamics or changes in leaf area that can feedback to leaf-gas exchange, albeit at longer time scales compared to the opening and closure of stomata in response to environmental stimuli (Buckley and Schymanski, 2014). Considering $\lambda$ as a

fitting parameter captures some trends in the data with respect to environmental stimuli such as vapor pressure deficit, temperature, or photosynthetically active radiation, but does not provide theoretical insights into stomatal responses to elevated $CO_2$. In a more theoretically complete approach, the stomatal optimization problem can be formulated to explicitly consider the impact of stomatal conductance on the dynamic nature of soil water—in other words, accounting for the

constraint that utilizing water quickly today necessarily reduces its availability tomorrow (Lu et al., in review). With this 'dynamic feedback' approach to stomatal optimization, $\lambda$ becomes an internal variable to be solved for in the optimization (Manzoni et al., 2013; Mrad et al., 2019). This 'dynamic feedback' approach considers soil water as a limited resource, but it can be further generalized by also considering the limitations on the transpiration rate imposed by reduced water transport from the soil to the leaves (Lu et al., 2020). The combined stomatal and leaf area responses to atmospheric $CO_2$ concentrations have not been explored with these three variants of stomatal optimization models, specifically i) 'instantaneous' optimization (OPT1); ii) 'dynamic feedback' optimization with no effect of water limitation in dry conditions (OPT2); and iii) 'dynamic feedback' optimization including the effect of water limitation in dry conditions (OPT3).

In this contribution, the PETA model and the three optimization model variants are compared, providing a set of predictions in the form of compact analytical equations. These equations, in turn, quantify the sensitivity of gas exchange rates (especially transpiration) to changing climatic conditions, and thus address the following questions:

1. How do physiological (stomatal conductance) and morphological (leaf area) adjustments coordinate to determine leaf and canopy gas exchange rates under atmospheric $CO_2$ concentrations?

2. How do these physiological and morphological adjustments vary under combined changes in $CO_2$ concentration and atmospheric or soil drought?

By comparing the predictions of the PETA and optimization models, a theoretical perspective on these questions is offered, while identifying advantages and limitations in these different modelling approaches.

## 2      Theory

Both PETA and optimization models describe leaf and canopy exchanges of water vapor and $CO_2$. They rest on three key simplifications. First, the entire canopy is subject to the same conditions and well-coupled to the atmosphere; i.e., the 'big leaf' approximation is used (Sect. 2.1). Second, plants are assumed to have reached an equilibrium at yearly to decadal time scales; i.e., they have acclimated to the atmospheric conditions by varying their growing season LAI (which is prescribed in both models) and stomatal conductance. At the shorter time scale of a dry-down, plants are assumed to maintain static leaf area, while they can still adjust stomatal conductance in response to variations in soil water. Third, photosynthetic capacity and vapour pressure deficit are considered fixed over the dry-down duration but allowed to vary at climatic time scales (in this way, they are treated as model parameters instead of dynamic or control variables). The models differ in the way stomatal responses are modelled (Figure 1; Sect. 2.2 and 2.3), but, to facilitate the model inter-comparison, the same dependence of LAI to atmospheric $CO_2$ concentration was considered. All symbols are defined in Table 1.

**Table 1. Definitions of symbols (including units), and subscripts and superscripts.**

| Symbol | Definition | Units |
|---|---|---|
| $a$ | Ratio of the diffusivities of $H_2O$ and $CO_2$ ($a=1.6$) | - |
| $a_1$ | Maximum Rubisco carboxylation capacity | $\mu$mol $CO_2$ (m$^2$ leaf)$^{-1}$ s$^{-1}$ |
| $a_2$ | Half saturation constant for net $CO_2$ assimilation | $\mu$mol $CO_2$ (mol air)$^{-1}$ |
| $A$ | Net canopy $CO_2$ assimilation rate | $\mu$mol $CO_2$ (m$^2$ ground)$^{-1}$ s$^{-1}$ |
| $A_L$ | Net leaf $CO_2$ assimilation rate | $\mu$mol $CO_2$ (m$^2$ leaf)$^{-1}$ s$^{-1}$ |
| $c_a$ | $CO_2$ concentration in the atmosphere | $\mu$mol $CO_2$ (mol air)$^{-1}$ |
| $D$ | Vapor pressure deficit | mol $H_2O$ (mol air)$^{-1}$ |
| $E$ | Canopy transpiration rate | mol $H_2O$ (m$^2$ ground)$^{-1}$ s$^{-1}$ |
| $E_L$ | Leaf transpiration rate | mol $H_2O$ (m$^2$ leaf)$^{-1}$ s$^{-1}$ |
| $E_w$ | Canopy transpiration rate under water supply-limited conditions | mol $H_2O$ (m$^2$ ground)$^{-1}$ s$^{-1}$ |
| $g$ | Stomatal conductance to $CO_2$ | mol air (m$^2$ leaf)$^{-1}$ s$^{-1}$ |
| $g_w$ | Stomatal conductance to $CO_2$ under water supply-limited conditions | mol air (m$^2$ leaf)$^{-1}$ s$^{-1}$ |
| $H$ | Hamiltonian ($H = A - \lambda E$) | $\mu$mol $CO_2$ (m$^2$ ground)$^{-1}$ s$^{-1}$ |
| $J$ | Canopy C gain over the period $T$ (objective function) | $\mu$mol $CO_2$ (m$^2$ ground)$^{-1}$ |
| $k$ | Carboxylation capacity ($= a_1/(a_2 + \chi c_a)$) | mol air (m$^2$ leaf)$^{-1}$ s$^{-1}$ |
| $L$ | Leaf area index | m$^2$ leaf (m$^2$ ground)$^{-1}$ |
| $M_w$ | Molecular weight of water ($M_w$=18 g (mol $H_2O$)$^{-1}$) | g (mol $H_2O$)$^{-1}$ |
| $x$ | Relative volumetric soil moisture (saturation normalized between wilting point and field capacity so $0\leq x \leq 1$) | - |
| $x_0$ | Initial relative volumetric soil moisture | - |
| $x_T$ | Final relative volumetric soil moisture | - |
| $T_a$ | Air temperature (assumed equal to canopy temperature) | °C |
| $t_d$ | Mean length of dry-down | d |
| $t_{day}$ | Daylight time conversion factor ($T_{day}$ =3600×12 s d$^{-1}$) | s d$^{-1}$ |
| $w_0$ | Root zone storage capacity | m |
| $Z_r$ | Rooting depth | m |
| $\alpha$ | Resource availability index | - |
| $\beta$ | Exponent of the rooting depth *vs.* leaf area relation ($Z_r \sim L^\beta$, see Appendix C) | - |
| $\chi$ | Ratio of internal to atmospheric $CO_2$ concentrations | - |
| $\Delta\varphi$ | Finite variation of the generic quantity $\varphi$ between future and current values | Same units as $\varphi$ |

| | | |
|---|---|---|
| $\kappa$ | Proportionality constant in the $E_w(x)$ relation | d$^{-1}$ |
| $\lambda$ | Lagrange multiplier | $\mu$mol $CO_2$ (mol $H_2O$)$^{-1}$ |
| $\nu$ | Unit conversion factor ($\nu = t_{day}M_w/\rho_w$) | m$^3$ s (mol $H_2O$)$^{-1}$ d$^{-1}$ |
| $\rho_w$ | Density of liquid water ($\rho_w = 10^6$ g m$^{-3}$) | g m$^{-3}$ |
| $\omega$ | Leaf or canopy water use efficiency ($\omega = A_L/E_L = A/E$) | $\mu$mol $CO_2$ (mol $H_2O$)$^{-1}$ |
| $\omega_i$ | Intrinsic leaf or canopy water use efficiency ($\omega_i = \omega D$) | $\mu$mol $CO_2$ (mol air)$^{-1}$ |
| Subscripts and superscripts | | |
| $t$ | Subscript indicating future conditions at a generic time $t$ | |
| $opt$ | Subscript indicating optimal stomatal conductance, transpiration rate, assimilation rate, or water use efficiency | |
| $w$ | Subscript indicating water-limited conditions | |
| * | Superscript indicating the transition point between well-watered and water-limited conditions | |
| $\bar{\varphi}$ | Overbar indicates temporal averaging of the generic quantity $\varphi$ (Eq. (14)) | |

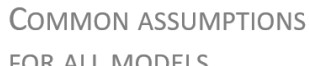

COMMON ASSUMPTIONS FOR ALL MODELS

Well-coupled canopy
Canopy fluxes ≈ leaf fluxes × LAI

$\alpha$: resource availability index
(high $\alpha$ → closed canopy
→ no $CO_2$ effect of leaf area)

OPTIMAL STOMATAL REGULATION

**PETA**

**OPT1**
Instantaneous optimization (unspecified $\lambda$)

Dynamic feedback ($\lambda$ calculated)

**OPT2**
water supply not limited

**OPT3**
water supply limited in dry conditions

SPECIFIC MODEL ASSUMPTIONS

$$\frac{\Delta E_L}{E_L}\left(\frac{\Delta c_a}{c_a},\frac{\Delta D}{D}\right)$$

$$\frac{\Delta A_L}{A_L}\left(\frac{\Delta c_a}{c_a},\frac{\Delta D}{D}\right)$$

- Soil water constraint not specified
- Time scale of optimization ~day

Soil water, $x$ — $x_0$ ... $x_T$ — 0

Supply not limited
Not available
All plant available water is used

Soil water, $x$ — $x_0$ ... $x_T$ — 0

Supply not limited
Supply limited
Not available
All plant available water is used

SPECIFIC MODEL RESULTS

OPT1, OPT2, OPT3 provide stomatal conductance and gas exchange rates as a function of $c_a, D, T_a, t_d$

$$\frac{\Delta E_{L,opt}}{E_{L,opt}}\left(\lambda,\frac{\Delta\varphi}{\varphi}\right)$$

$$\frac{\Delta A_{L,opt}}{A_{L,opt}}\left(\lambda,\frac{\Delta\varphi}{\varphi}\right)$$

$$\frac{\Delta E_{L,opt}}{E_{L,opt}}\left(\frac{\Delta\varphi}{\varphi}\right)$$

$$\frac{\Delta A_{L,opt}}{A_{L,opt}}\left(\frac{\Delta\varphi}{\varphi}\right)$$

$$\frac{\Delta\overline{E_{L,opt}}}{\overline{E_{L,opt}}}\left(\frac{\Delta\varphi}{\varphi}\right)$$

$$\frac{\Delta\overline{A_{L,opt}}}{\overline{A_{L,opt}}}\left(\frac{\Delta\varphi}{\varphi}\right)$$

RESULTS FROM ALL MODELS

$$\frac{\Delta E}{E},\frac{\Delta A}{A},\frac{\Delta\omega}{\omega} = \text{functions of variations in } \varphi = c_a, D, T_a, t_d$$

**Fig. 1. Conceptual representation of the PETA and stomatal optimization models used to assess gas exchange responses (transpiration $E$ and net $CO_2$ assimilation $A$) to changes in atmospheric $CO_2$ concentrations $c_a$, vapor pressure deficit $D$ (either independent of or caused by changes in air temperature $T_a$), and length of a representative dry-down $t_d$ (during which soil moisture $x$ decreases from the initial value $x_0$ to $x_T$). Three variants of the stomatal optimization model are considered: i) 'instantaneous' optimization (OPT1, where the marginal water use efficiency $\lambda$ is unspecified), ii) 'dynamic feedback' optimization with no effect of water limitation in dry conditions (OPT2), and iii) 'dynamic feedback' optimization including the effect of water limitation in the 'supply limited' regime (OPT3). In the heuristic PETA model, leaf-level gas exchange responses (subscript $L$) follow from the empirical relation between water use efficiency ($\omega = A/E$) and $c_a$ and $D$, whereas they are results of optimal stomatal regulation in the optimization models (subscript $opt$). Overbar indicates temporal averaging during a representative dry-down period; $\varphi$ indicates a generic climatic variable ($c_a$, $D$, $T_a$, or $t_d$).**

## 2.1 Leaf- and canopy-level transpiration and assimilation rates

Leaf-level transpiration rate $E_L$ (mol $H_2O$ ($m^2$ leaf)$^{-1}$ s$^{-1}$) and leaf $CO_2$ uptake rate $A_L$ (µmol $CO_2$ ($m^2$ leaf)$^{-1}$ s$^{-1}$) are described as diffusion-driven processes with negligible leaf boundary layer resistance,

$$E_L = agD,$$
$$A_l = g(c_a - c_i), \tag{1}$$

where in the first equation, $a$=1.6 is the ratio between the diffusivities of water vapour and $CO_2$ (nondimensional), $g$ is the stomatal conductance to $CO_2$ (mol air ($m^2$ leaf)$^{-1}$ s$^{-1}$), and $D$ is the atmospheric vapor pressure deficit expressed as a molar fraction (mol $H_2O$ (mol air)$^{-1}$). In the second equation, $A_L$ is described as a $CO_2$ flux mediated by $g$ and driven by the difference between atmospheric and leaf internal $CO_2$ concentrations (respectively $c_a$ and $c_i$, expressed in µmol $CO_2$ (mol air)$^{-1}$). Mass conservation further implies that the rate of $CO_2$ uptake into the leaf must equal the net $CO_2$ assimilation rate. The net assimilation rate can be modelled as a function of internal $CO_2$ concentration as

$$A_L = \frac{a_1 c_i}{a_2 + c_i} \approx \frac{a_1 c_i}{a_2 + \chi c_a} = k c_i, \tag{2}$$

where $a_1$ and $a_2$ are temperature-dependent kinetic constants that we assume are independent of $c_a$ as a first approximation, and $k$ is the maximum Rubisco carboxylation capacity (mol air ($m^2$ leaf)$^{-1}$ s$^{-1}$). The parameters defining $k$ can be related to light availability and temperature, but we assume here that light is fixed and long-term mean temperature is varied as a model parameter. Following Katul et al. (2010), $c_i$ in the denominator of the second term is approximated as $c_i \approx \chi c_a$, where $\chi$ is the long-term ratio of leaf internal to atmospheric $CO_2$ concentration, so that $k = a_1/(a_2 + \chi c_a)$. This assumption is reasonable when $a_2$ is commensurate to or larger than $c_i$ (which is expected for Rubisco limited assimilation), so that variations in $c_i$ can be ignored when summed to $a_2$ in the denominator of the first equality. As a result, $A_L$ is a linear function of $c_i$, but the slope of the relation decreases with increasing atmospheric $CO_2$ concentration. Moreover, this approximation allows retaining variations in $c_i/c_a$ with $c_a$ (Katul et al., 2010). Equating the rates of $CO_2$ uptake and assimilation yields a relation between $A_L$ and $g$ (e.g., Hari et al., 1986),

$$A_L = \frac{gk}{g+k} c_a. \tag{3}$$

Therefore, increasing atmospheric $CO_2$ concentration affects the net $CO_2$ assimilation rate via two direct effects—it increases the available $CO_2$ in the leaf (through $c_a$) and it decreases the marginal return on $CO_2$ fixation at high $CO_2$ concentrations (through $k$). Temperature effects on $k$ are considered using the temperature response functions for Rubisco-limited assimilation of Medlyn et al. (2002). While $A_L$ is described by Eq. (3) in the three variants of the optimization model, in the PETA model, the response of $A_L$ to environmental variations are described based on heuristic arguments that combine water and $CO_2$ fluxes from Eq. (1) (Sect. 2.2).

Compared to the equations above, nonlinear models of assimilation accounting for Rubisco or RuBP regeneration limitation (Farquhar et al., 1980; Vico et al., 2013; Katul et al., 2010) would yield a more complex relation between $A_L$ and $g$. These

170 complex relations allow exploring short-term responses of gas exchange to variations in temperature, VPD, and photosynthetically active radiation (Medlyn et al., 2011; Katul et al., 2010; Vico et al., 2013). However, here we focus on long-term responses to $CO_2$ concentration, which are not affected by the specific choice of assimilation kinetics, as demonstrated in the following. We thus select the simplest model for $A_L$ for the sake of mathematical tractability.

Further assuming the big-leaf approximation and that the canopy is well-coupled with the atmosphere, the canopy-level
transpiration ($E$) and $CO_2$ assimilation rates ($A$), can be estimated as the leaf-level exchange scaled up by the LAI ($L$)

$$E = E_L L,$$
$$A = A_L L. \tag{4}$$

Hence, by promoting plant growth and larger LAI, elevated atmospheric $CO_2$ levels can have an indirect effect on gas exchange mediated by $L$—in addition to any direct effects on $g$ or $A_L$. This linear scaling does not capture nonlinear effects of leaf area on $CO_2$ uptake, such as decreasing returns of higher LAI due to self-shading and redistribution of nitrogen (dePury and Farquhar, 1997). It also neglects the effect of aerodynamic resistance on canopy gas exchange, which can be
large in dense canopies (Juang et al., 2008). However, this simplification does not strongly affect the sensitivity of gas exchange rates to changes in atmospheric $CO_2$ concentrations (Donohue et al., 2017). Therefore, we expect that the consequences of increasing LAI on gas exchange could be magnified at high LAI values with this model, though this effect should be relatively small.

Knowing transpiration and $CO_2$ assimilation rates, the instantaneous water use efficiency (WUE) is given as $\omega = A_L/E_L = $
$A/E$. The intrinsic water use efficiency (i.e., the ratio of net $CO_2$ assimilation rate and stomatal conductance) is linked to $\omega$ as $\omega_i = \omega D$. Due to the linear scaling from leaf- to canopy-levels, both WUE and intrinsic WUE are numerically the same at these two spatial scales.

## 2.2 Partitioning of Equilibrium Transpiration and Assimilation (PETA) model

The PETA model is formulated as a set of relations between the relative changes of variables related to leaf gas exchange and the relative change in atmospheric $CO_2$ concentration and VPD. In Donohue et al. (2013, 2017), the premise of PETA is that leaf-level WUE ($\omega$) scales linearly with $c_a$ (see also Lavergne et al., 2019), and inversely with the square root of VPD. This relation can be explained by the definition of WUE using Eq. (1) for $A_L$ and $E_L$; i.e., $\omega = A_L/E_L \sim c_a(1 - \chi)/D$, where $\chi$ decreases with increasing $D$ as a result of stomatal closure while photosynthesis continues, leading to $\omega \sim c_a/\sqrt{D}$
(Donohue et al., 2013 and references therein). The relative change in $\omega$ depends, by definition, on $A_L$ and $E_L$, and thus also on $c_a$ and $D$ according to the following relations (Donohue et al., 2017),

$$\frac{\Delta\omega}{\omega} = \frac{1+\frac{\Delta A_L}{A_L}}{1+\frac{\Delta E_L}{E_L}} - 1 \approx \frac{1+\frac{\Delta c_a}{c_a}}{1+\frac{\Delta\sqrt{D}}{\sqrt{D}}} - 1 = \frac{1+\frac{\Delta c_a}{c_a}}{\sqrt{1+\frac{\Delta D}{D}}} - 1. \tag{5}$$

In Eq. (5) and in the following, the symbol $\Delta$ indicates a finite (not infinitesimal) variation; i.e., the value at a future time $t$ minus the current time value (e.g., $\Delta c_a = c_{a,t} - c_a$). The equality on the far right-hand side of Eq. (5) is obtained by noting that $\Delta\sqrt{D}/\sqrt{D} = \sqrt{1 + \Delta D/D} - 1$, which allows expressing the variation in $\omega$ as a function of the relative variation in $D$ rather than the variation of its square root. The PETA model then links heuristically the expected relative changes in $L$, $A_L$, and $E_L$ to changes in $\omega$ as driven by $c_a$ and $D$, and to 'resource availability' as quantified by an index $\alpha$ ($0 \leq \alpha \leq 1$). This index represents how far vegetation is from the maximum $L$ expected for that location. High $\alpha$ indicates an old stand or in general a stand with $L$ close to the maximum, where additional leaf area increases are not possible (see also Sect. 2.5). With these premises, the relative changes are expressed heuristically in the PETA model as (Donohue et al., 2017)

$$\frac{\Delta L}{L} = \frac{\Delta\omega}{\omega}(1-\alpha)^2,$$

$$\frac{\Delta A_L}{A_L} = \frac{\Delta\omega}{\omega}\alpha,$$

$$\frac{\Delta E_L}{E_L} = \left(\frac{1}{1+\frac{\Delta\omega}{\omega}} - 1\right)(1-\alpha). \tag{6}$$

When changes in $D$ are small, and variations in WUE are mostly driven by $c_a$, Eq. (5) reduces to $\Delta\omega/\omega \approx \Delta c_a/c_a$, and the variations in $L$, $A_L$, and $E_L$ can be recalculated accordingly. The relations between leaf area and gas exchange rates with $c_a$ implicit in Eq. (6) can be explained as follows:

- In an open canopy far from the maximum $L$ for that site (i.e., $\alpha \to 0$), increases in $c_a$ allow higher leaf area ($\Delta L/L \to \Delta\omega/\omega$), while $CO_2$ assimilation rate per leaf area remains unchanged ($\Delta A_L/A_L \to 0$), and transpiration rate per leaf area decreases (i.e., $c_a$ causes a structural response compensated for by stomatal closure at the leaf level).

- In a closed canopy (i.e., $\alpha \to 1$), increases in $c_a$ do not cause changes in leaf area, which is already near the maximum value for that site ($\Delta L/L \to 0$); however, net assimilation rate per leaf area increases ($\Delta A_L/A_L \to \Delta\omega/\omega$), while transpiration rate per leaf area remains unchanged ($\Delta E_L/E_L \to 0$).

The relations between relative changes in canopy transpiration and photosynthesis and changes in $c_a$ are found by multiplying the leaf-level fluxes by $L$ (Eq. (4)), obtaining

$$\frac{\Delta A}{A} = \left(1 + \frac{\Delta A_L}{A_L}\right)\left(1 + \frac{\Delta L}{L}\right) - 1,$$

$$\frac{\Delta E}{E} = \left(1 + \frac{\Delta E_L}{E_L}\right)\left(1 + \frac{\Delta L}{L}\right) - 1. \tag{7}$$

Equations (6) and (7) link the changes in gas exchange rates to the changes in atmospheric $CO_2$ concentration for a given canopy status as represented by $\alpha$. Equation (7) also shows that canopy transpiration can vary unless both leaf-level transpiration and leaf area index are constant. This result of the PETA model differs from a key assumption of the stomatal optimization model (Sections 2.3.2 and 2.3.3). Finally, we can calculate the variation in intrinsic WUE ($\omega_i = \omega/D$),

$$\frac{\Delta\omega_i}{\omega_i} = \left(1 + \frac{\Delta\omega}{\omega}\right)\left(1 + \frac{\Delta D}{D}\right) - 1. \tag{8}$$

A simplified version of the PETA model is described in Appendix A and used to develop analytical arguments in the Discussion.

## 2.3  Optimal stomatal control models

The optimal stomatal conductance model is formulated as an optimal control problem with the objective to maximize net $CO_2$ assimilation at the canopy level over a set time interval $t_d$ (duration of a representative dry period), subject to the

constraint that soil moisture $x$ is limited. This model also assumes that plants, over a period much longer than $t_d$, can alter allocation and thus leaf area in response to atmospheric $CO_2$ concentration (as in the PETA model). Detailed mathematical derivations are provided in Appendix B. Here we report only the equations for optimal stomatal conductance, based on which all gas exchange rates can be calculated (Eq. (1), (3), and (4)). Solving the optimization problem involves the calculation of the Lagrange multiplier ($\lambda$), an auxiliary variable that accounts for the soil moisture constraint and that can be

interpreted as the marginal water use efficiency. Three different analytical equations for the optimal $g$ are obtained depending on the specific assumptions made when setting up the optimization problem: i) instantaneous optimization where $\lambda$ is treated as a fitting parameter (OPT1), ii) dynamic feedback optimization where $\lambda$ is derived mathematically before obtaining the optimal stomatal conductance, but where transpiration is independent of soil moisture until the available water has been consumed (OPT2), and iii) dynamic feedback optimization where transpiration is reduced as soil dries (OPT3).

In versions OPT2 and OPT3, a model of soil moisture dynamics needs to be added to the gas exchange equations. Neglecting evaporation from the soil or canopy surface, the soil water balance during a dry-down with negligible precipitation can be written as (in units of m d$^{-1}$),

$$w_0 \frac{dx}{dt} = -\nu E, \text{ with initial condition } x(0) = x_0, \tag{9}$$

where $x$ is the plant-available relative soil moisture (i.e., the saturation level rescaled between 0 at the wilting point and 1 at field capacity, as in Porporato et al., 2004), $w_0$ is the root zone water storage capacity (m), $\nu$ is a unit conversion factor to

make the units of $E$ in Eq. (4) (mol H$_2$O (m$^2$ ground)$^{-1}$ s$^{-1}$) consistent with typical units used in water balance equations (m d$^{-1}$): $\nu = t_{day} M_w / \rho_w$ (m$^3$ s (mol H$_2$O)$^{-1}$ d$^{-1}$), with $t_{day} = 3600 \times 12$ s d$^{-1}$: active transpiration period in a day, $M_w = 18$ g (mol H$_2$O)$^{-1}$: molecular weight of water; $\rho_w = 10^6$ g m$^{-3}$: density of liquid water. The dry-down starts at a soil moisture $x_0$ below field capacity, so that the only water loss from the soil in Eq. (9) is $E$, and lasts for a period $T$, leaving a residual amount of water $x_T$ at the end.

### 2.3.1    OPT1: instantaneous stomatal optimization

If stomatal conductance is allowed to vary through time but independently of soil moisture, the Lagrange multiplier of the optimization is time-invariant. Substituting Eq. (1) and (3) in Eq. (24) in Appendix B1 and solving for $g$ yields (Hari et al., 1986; Katul et al., 2010; Lloyd and Farquhar, 1994; Palmroth et al., 1999)

$$g_{opt} = k \left( \sqrt{\frac{c_a}{a\lambda D}} - 1 \right), \tag{10}$$

where $\lambda$ is regarded as an adjustable parameter. Because the effects of soil moisture dynamics on stomatal conductance are
neglected, this approach is termed 'instantaneous' optimization. For a set value of $\lambda$, Eq. (10) describes the short-term
responses of stomatal conductance to $c_a$, $D$, and any environmental condition affecting $k$. However, this equation neglects the
fact that soil water is limited; i.e., no constraints are imposed on how much water can be transpired in a given time interval.

### 2.3.2    OPT2: dynamic feedback optimization with transpiration rate independent of soil moisture

A more realistic approach that overcomes the limitation of a freely adjustable $\lambda$ is determining the value of $\lambda$ by imposing
the constraint that the initial soil moisture $x_0$ is depleted, leaving only $x_T$ at the end of the time interval $t_d$. This constraint
provides an additional equation that allows us to determine $\lambda$ (Eq. (25) in Appendix B1). Thus, $\lambda$ in OPT2 is not simply an
adjustable parameter (as it has been treated previously), but rather a clearly defined property of the coupled soil-plant
system, including the ending soil moisture and the duration of the dry period. With the obtained $\lambda$, the optimal stomatal
conductance is found as (solid line in Fig. 2a),

$$g_{opt} = \frac{w_0(x_0 - x_T)}{vaDLt_d}, \tag{11}$$

which shows that stomatal conductance (and thus also transpiration and net $CO_2$ assimilation rates) is independent of time or
soil moisture but vary with soil water storage capacity, $w_0(x_0 - x_T)$, and other environmental conditions (recall that $c_a$, $D$,
and $k$ are time invariant during the dry-down, but allowed to vary at longer time scales over which climatic changes occur).
It is important to emphasize that this specific stomatal conductance trajectory is not a result of our assumption that all
available water is used. Rather, it is the solution that best balances the water consumption rate over time to maximize net
assimilation. Even without a direct dependence of gas exchange on soil moisture (which is explored in OPT3), this solution
accounts for soil moisture dynamics, because faster transpiration reduces soil water storage more rapidly. In this sense, this
approach is denoted 'dynamic feedback' optimization.

Equation (11) could be also found by simply imposing that the stomatal conductance adjusts to use all the water in the
allotted time (details are shown in Sect. 3.1). Therefore, assuming optimal stomatal control and a finite amount of plant-
available water results in a stomatal conductance equation that is independent of the atmospheric $CO_2$ concentration (no
direct control), but that is inversely proportional to LAI. This implies an inverse, indirect control of atmospheric $CO_2$
concentration on leaf-level stomatal conductance. In contrast, leaf-level net $CO_2$ assimilation rate increases with atmospheric
$CO_2$ concentration (direct control), even though this effect decreases at high $c_a$ due to the dependence of $k$ on $c_a$ (in Eq. (2)).
The canopy-level optimal stomatal conductance and $CO_2$ assimilation rate are simply obtained from the leaf-level quantities
using Eq. (4).

The equations of OPT2 can be used in two ways. Environmental conditions and soil parameters can be set to the long-term
mean values and $\lambda$ determined accordingly with Eq. (26) in Appendix B1; the same mean conditions can be used in Eq. (11)

(in combination with the equations for transpiration and net assimilation rates) to study the responses of gas exchange to long-term climatic changes. This is the approach we will follow in this contribution. Alternatively, one can calculate $\lambda$ based

on the long-term mean environmental conditions and soil parameters, insert that specific value in Eq. (10), and then study the short-term responses of stomatal conductance to changes in $c_a$, $D$, and $k$ for given $\lambda$. This solution still accounts for the dynamic feedback mechanism, but allows studying responses to fluctuations around the long-term mean conditions as captured by the value of $\lambda$.

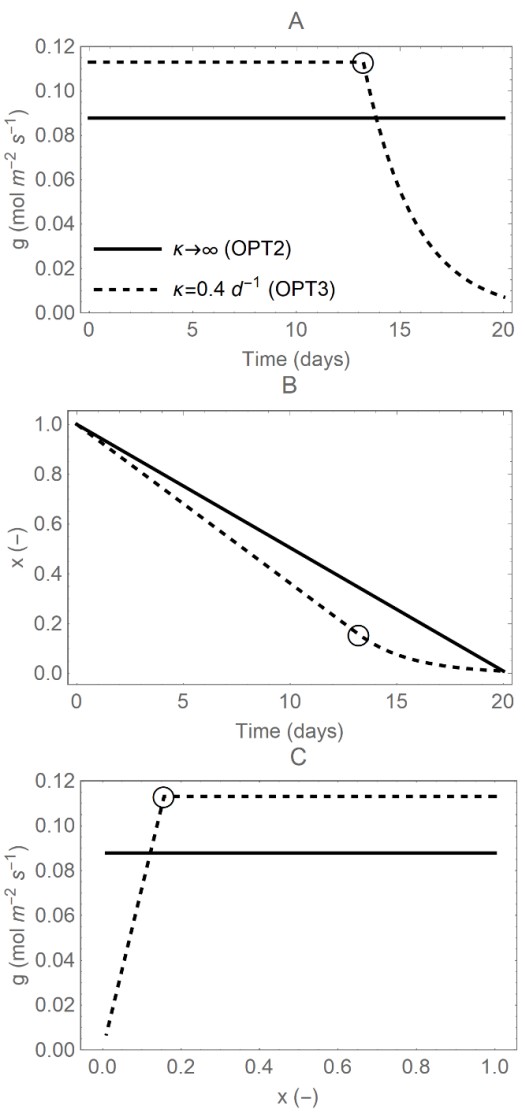

**Fig. 2. Temporal trajectories of A) leaf-level stomatal conductance ($g$), B) plant available soil moisture ($x$), and C) relations between $g$ and $x$ (with time increasing from right to left), during a single dry period of duration $t_d$=20 d. Line styles indicate when water supply from the soil is unlimited (OPT2: solid line, infinite $\kappa$; Section 2.3.2) or limited in dry conditions (OPT3: dashed line, finite $\kappa$; Section 2.3.3). Open circles indicate the transition points to water limited conditions ($x^*$ and $g_{opt}^*$ at time $t^*$; see details in Appendix B). Parameter values are as in Table 2.**

### 2.3.3    OPT3: dynamic feedback optimization with transpiration rate limited by soil moisture

Different from OPT1 and OPT2, we now consider soil moisture limitations on gas exchange (dashed lines in Fig. 2). Stomatal conductance is reduced as soil moisture decreases during a dry period, because of the combined effect of lowered water pressures along the soil-plant system and reduced conductance to water transport in the soil and the plant xylem

(Cruiziat et al., 2002; Klein, 2014). As a result, transpiration rate proceeds at a high and stable rate in well-watered conditions, but decreases approximately linearly as soil moisture declines due to stomatal closure and limited water supply from the soil (Sadras and Milroy, 1996). Based on this assumption, stomatal conductance decreases linearly with $x$ in dry conditions (i.e., late in the dry down, after a threshold time denoted by $t^*$; dashed line at low $x$ in Fig. 2c),

$$g_w = \frac{w_0 \kappa}{vaDL} x \text{ for } t > t^*. \tag{12}$$

In contrast, in well-watered conditions, stomatal conductance can be optimized. The optimal stomatal conductance is
calculated with Eq. (10) after finding the Lagrange multiplier specific for model OPT3, which differs from that in OPT2 because the boundary conditions of the optimization have changed. Therefore, when the soil is relatively moist, optimal stomatal conductance is found with an equation similar to OPT2, but modified to accounts for the fact that stomatal conductance will become water limited when $t > t^*$ (dashed line at high $x$ in Fig. 2c),

$$g_{opt} = \frac{x_0 w_0 \kappa}{vaDL(1 + \kappa t^*)} \text{ for } t \leq t^*. \tag{13}$$

The specific value of $t^*$ is determined as explained in Appendix B.

Predictions of the OPT3 model are functions of time and must be interpreted as time series, different from the time-invariant gas exchange rates of the other models (OPT1, OPT2, and PETA). Thus, to compare results to those from the other models, the time-averaged gas exchange rates are calculated as

$$\bar{\varphi} = \frac{\int_0^{t_d} \varphi(t) dt}{t_d}, \tag{14}$$

where $\varphi$ is used to represent any of the gas exchange variables ($E_L, A_L, E, A$) and the overbar indicates temporal averaging.

### 2.4  Comparing the results of optimization and PETA models

To compare the results of the optimization models with those of the PETA model, the relative changes in leaf transpiration and assimilation rates are calculated as

$$\frac{\Delta E_{L,opt}}{E_{L,opt}} = \frac{E_{L,opt,t}}{E_{L,opt}} - 1,$$

$$\frac{\Delta A_{L,opt}}{A_{L,opt}} = \frac{A_{L,opt,t}}{A_{L,opt}} - 1, \tag{15}$$

where $E_{L,opt}$ and $A_{L,opt}$ are evaluated at baseline (current) environmental conditions, and subscript $t$ indicates future climatic conditions. To make the equations of the PETA and optimization models comparable, future values of $c_a$, $D$, $L$, and $t_d$ appearing in the equations for the optimal gas exchange rates are expressed as $c_{a,t} = (\Delta c_a/c_a + 1)c_a$, $D_t = (\Delta D/D + 1)D$, $L_t = (\Delta L/L + 1)L$, and $t_{d,t} = (\Delta t_d/t_d + 1)t_d$. Furthermore, the same LAI changes are included in both PETA and optimization models by combining Eq. (5) and (6) to determine $\Delta L/L$. Leaf-level rates in the optimization model variants are scaled up to the canopy-level as in the PETA model (Eq. (7)), thus including the additional indirect effect of atmospheric $CO_2$ concentration on LAI.

The relative changes for transpiration can be re-written in a compact form at both the leaf and canopy levels for OPT2 and OPT3 (after some algebraic manipulation of Eq. (1), (4), and (11)),

$$\frac{\Delta E_{L,opt}}{E_{L,opt}} = \frac{1}{\left(\frac{\Delta L}{L}+1\right)\left(\frac{\Delta t_d}{t_d}+1\right)} - 1,$$

$$\frac{\Delta E_{opt}}{E_{opt}} = \frac{1}{\frac{\Delta t_d}{t_d}+1} - 1. \tag{16}$$

While in the PETA model the water use efficiency $\omega$ is prescribed (Eq. (5)), in the optimization model $\omega$ is obtained as a result of the optimization, $\omega_{opt} = \frac{A_{L,opt}}{E_{L,opt}} = \frac{A_{opt}}{E_{opt}}$. Accordingly, variations in $\omega$ in the optimization model induced by changing $CO_2$ concentration and VPD are calculated as,

$$\frac{\Delta \omega_{opt}}{\omega_{opt}} = \frac{\omega_{opt,t}}{\omega_{opt}} - 1. \tag{17}$$

Similarly, the variations in intrinsic water use efficiency are found using the definition $\omega_i = \omega D$ as,

$$\frac{\Delta \omega_{i,opt}}{\omega_{i,opt}} = \frac{\omega_{opt,t}D_t}{\omega_{opt}D} - 1. \tag{18}$$

In scenarios in which VPD does not change in the future (i.e., $D_t=D$), the variations in WUE and intrinsic WUE are the same.

## 2.5 Model parameters and climate change scenarios

The models are parameterized for a generic vegetation type and a baseline climate (Table 2), from which variations in gas exchanges for a wide range of future climate conditions are evaluated. In both the PETA and optimization models, LAI varies with atmospheric $CO_2$ concentration and VPD in the same manner (top of Fig. 1). Growth chamber and FACE experiments showed that LAI generally increases in open canopies and young stands with increasing atmospheric $CO_2$ concentration across plant functional types (symbols in Fig. 3). However, the rate of increase varies depending on growth conditions, with the LAI of closed-canopy and older plant communities responding less to increasing $CO_2$ levels than those of younger communities (Bader et al., 2013; Duursma et al., 2016). We test these effects by varying the parameter $\alpha$ (Donohue et al., 2017, 2013), which increases from zero, when leaf area responds the most to increasing $CO_2$ concentration (open canopy with low leaf area index and/or young plants), to one when leaf area is unresponsive (closed canopy with high

leaf area index and/or older plants). The intermediate value $\alpha =0.5$ is selected for the analyses involving simultaneous changes of atmospheric $CO_2$ concentration, VPD, and length of the dry period.

In the PETA model, $\alpha$ is the only adjustable parameter, so no further parameter selection is necessary. In the optimization model, we selected parameter values representative of $A$-$c_i$ curves for $C_3$ plants (Table 2). Soil parameters determining the water storage capacity $w_0$ is selected for a loamy soil and intermediate rooting depth (Table 2.1 in Rodriguez-Iturbe and Porporato, 2004). The baseline values of $c_a$, $D$ and $t_d$ represent current climatic conditions under a mild temperature regime. The assumed dry-down length of $t_d$=20 d corresponds to a dry spell length for which vegetation is adapted; i.e., $t_d$ is interpreted as a characteristic time between the length of the average dry period and that of an actual drought that would cause irreversible damage or mortality. The baseline $L$=2 $m^2$ $m^{-2}$ is reasonable for a relatively open canopy, meeting the assumption of well coupled conditions.

The $c_a$, $t_d$, and $D$ are allowed to vary in the ranges expected under future climatic conditions. We explore a range of $c_a$ from 400 to 800 µmol $CO_2$ (mol air)$^{-1}$ (maximum $\Delta c_a/c_a =1$), in line with atmospheric $CO_2$ concentration being expected to approximately double from 2016 to 2100 according to an intermediate-emission scenario (SSP3-7.0, IPCC, 2021).

The VPD can be changed by letting relative humidity vary at constant temperature or by letting temperature vary at constant relative humidity. The first scenario allows isolating the effect of VPD on stomatal conductance and transpiration alone. In the second scenario, VPD affects both water and $CO_2$ exchanges because of direct effects on the former and indirect effects on the latter via photosynthetic capacity (Medlyn et al., 2002), which in turn also affects gas exchange in the optimization models (again via $k$). To compare between the two scenarios, VPD is varied in the same range, even though projected variations in VPD are mostly attributed to warming (relative humidity variations are expected to be moderate). Taking the United States as an example, VPD is expected to increase between ~40 and ~65% by the end of the century, depending on the general circulation model used for the projections, with a median of ~50% (Ficklin and Novick, 2017; Yuan et al., 2019). While this value is probably higher than the global average, we use it as an upper bound for our sensitivity analyses (maximum $\Delta D/D =0.5$).

Dry period lengths during the growing season have been shifting towards either longer or shorter lengths depending on location, with historical variations up to $\sim\pm10\%$ per decade (Breinl et al., 2020). Because of this large variability in historical times, and the large uncertainty in projected dry period durations, we consider $t_d$ variations between $\pm50\%$ ($\Delta t_d/t_d$ ranges from -0.5 to +0.5).

**Table 2. Baseline parameter values (relative variations in $c_a$, $D$, $T_a$, and $L$ are calculated with respect to the values reported here).**

| Symbol | Value | Units | Notes and sources |
|--------|-------|-------|-------------------|
| $a_1$ | 100 | $\mu$mol $CO_2$ m$^{-2}$ s$^{-1}$ | Typical of C3 plants (Campbell and Norman, 1998) |
| $a_2$ | 710 | $\mu$mol $CO_2$ (mol air)$^{-1}$ | Typical of C3 plants (calculated after Medlyn et al., 2002) |
| $c_a$ | 410 | $\mu$mol $CO_2$ (mol air)$^{-1}$ | Ambient atmospheric $CO_2$ concentration in 2019 |
| $D$ | 0.015 | mol $H_2O$ (mol air)$^{-1}$ | Calculated at $T_a$=20 °C with 35% relative humidity |
| $L$ | 2 | m$^2$ m$^{-2}$ | Chosen value |
| $x_0$ | 1 | - | Equivalent to the field capacity |
| $x_T$ | 0.01 | - | Equivalent to the wilting point |
| $T_a$ | 20 | °C | Chosen value |
| $t_d$ | 20 | d | Chosen value |
| $w_0$ | 0.09 | m | Product of porosity (0.45 m$^3$ m$^{-3}$), rooting depth ($Z_r$=0.4 m), and difference in saturation between field capacity and wilting point (0.41 m$^3$ m$^{-3}$) for a loamy soil (Table 2.1 in Rodriguez-Iturbe and Porporato, 2004) |
| $Z_r$ | 0.4 | m | Chosen value; see Appendix C for details on how $Z_r$ relates to $L$ |
| $\alpha$ | 0.5 | - | Chosen value (intermediate leaf area allowing some degree of adjustment) |
| $\chi$ | 0.7 | - | Typical of C3 plants (Campbell and Norman, 1998) |
| $\kappa$ | 0.4 | d$^{-1}$ | Chosen value |

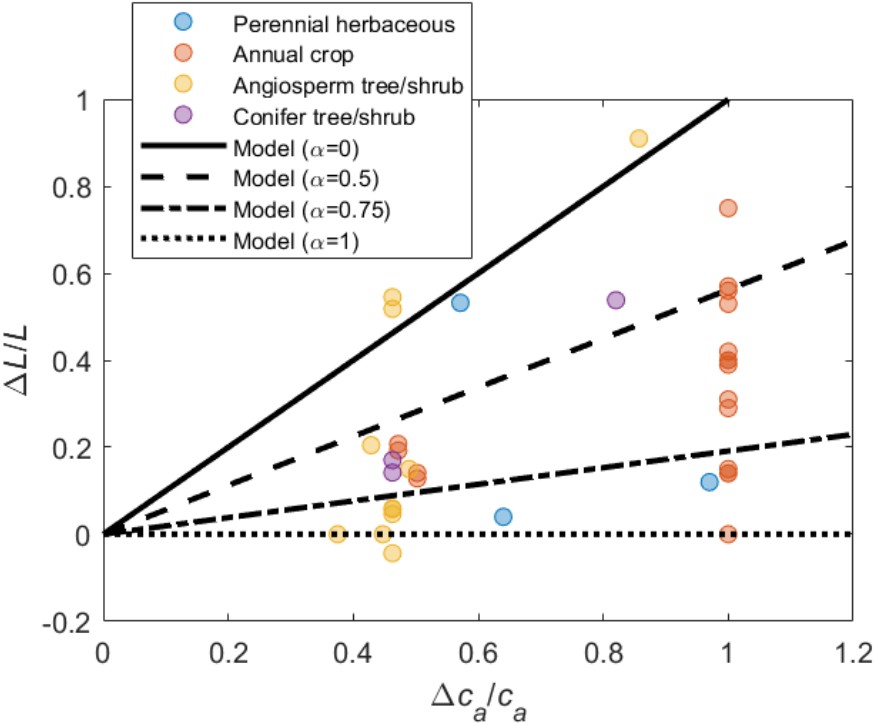

**Fig. 3.** Relative change in leaf area ($\Delta L/L$) as a function of relative change in atmospheric CO₂ concentration ($\Delta c_a/c_a$), across plant functional types (colours); lines show how the change in leaf area is modelled depending on the canopy status, indicated by $\alpha$ (higher $\alpha$ implies larger leaf area under ambient conditions and therefore lower sensitivity to changes in $c_a$, Eq. (6)). The effect of variations in vapor pressure deficit on leaf area are not considered in this figure, so that $\Delta L/L = \Delta c_a/c_a\,(1-\alpha)^2$. The same variations in $L$ due to $c_a$ (for given $\alpha$) are prescribed in both PETA and optimization models. Data points represent temporal averages of leaf area changes in response to elevated $c_a$ at plant to stand scales, shown to illustrate the range of observed responses (data and sources are reported in the Supplementary Information).

## 3    Results

We start by comparing the effects of atmospheric CO₂ concentration on gas exchange in the three variants of the optimization model (Fig. 4). Next, the CO₂ effects are assessed in both the PETA and optimization models at fixed VPD, but with different values of $\alpha$ (Fig. 5). Finally, the combined effects of CO₂ concentration and VPD (Fig. 6-7), and CO₂ concentration and dry period length (Fig. 8) are assessed in both models. An additional analysis is conducted in Appendix C to test how a coordinated deepening of the roots and increased leaf area index could affect the gas exchange sensitivity to elevated CO₂.

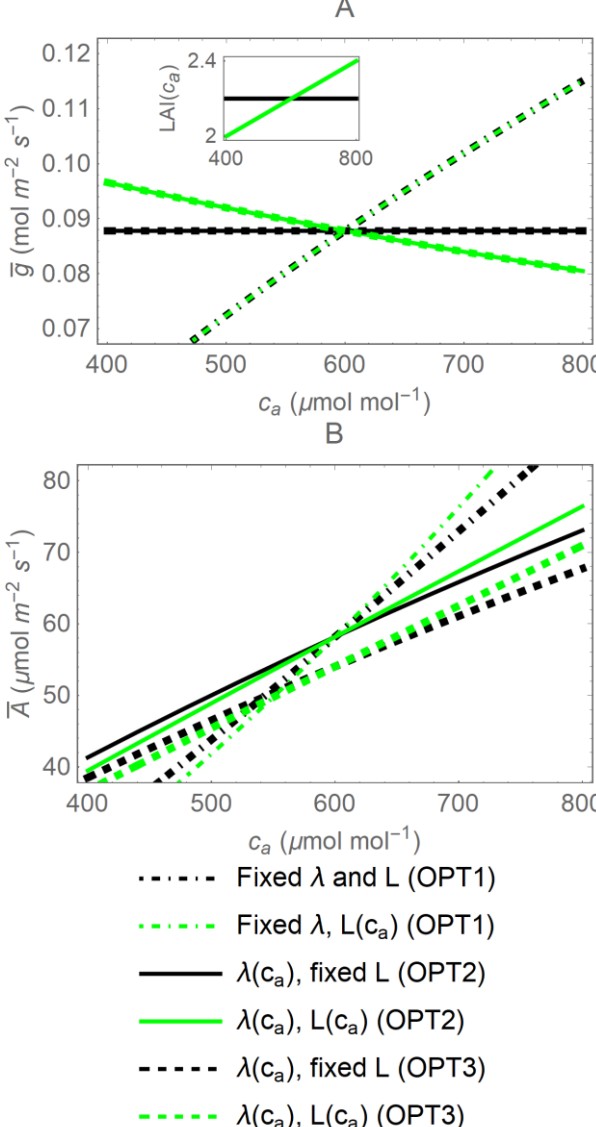

**Fig. 4.** Effect of relative plant available soil moisture ($x$) and atmospheric $CO_2$ concentration ($c_a$) on gas exchange as predicted by three variants of the stomatal optimization model (identified by different line dashing). A) Mean stomatal conductance ($\bar{g}$) and B) mean canopy net $CO_2$ assimilation rate ($\bar{A}$) during a dry period of $T_d$=20 d as a function of $c_a$, when transpiration is either independent of soil moisture (OPT2, solid lines) or water limited in dry conditions (OPT3, dashed lines), and with leaf area index ($L$) acclimating with increasing $c_a$ or fixed (green *vs.* black lines, respectively). The dot-dashed lines refer to the 'instantaneous' optimal stomatal conductance (OPT1), obtained from Eq. (10) with $\lambda$ set to a constant value (Eq. (25) at $c_a$=600 μmol $CO_2$ (mol air)$^{-1}$). Note that lines of different thickness are used to distinguish overlapping curves. The inset in panel A shows how $L$ varies with $c_a$; to make visual comparisons easier, $L$ variations are centred around a common value for all model variants at $c_a$=600 μmol $CO_2$ (mol air)$^{-1}$. Parameter values are as in Table 2.

## 3.1 Optimal stomatal conductance under varying atmospheric CO₂ concentration

Different variants of the optimization model predict contrasting responses to atmospheric $CO_2$ concentration. The instantaneous optimization OPT1 (in which $\lambda$ is a fixed parameter, Eq. (10)) predicts increasing stomatal conductance with increasing $c_a$ regardless of LAI (black and green dot dashed lines in Fig. 4A). Conversely, with increasing $c_a$, the dynamic feedback optimization OPT2 (Eq. (11)) predicts that stomatal conductance is stable when LAI is fixed or decreasing when LAI acclimates with $c_a$ (solid black and green lines in Fig. 4A, respectively).

The mean stomatal conductance ($\bar{g}$) over the dry-down is independent of whether soil water becomes limiting or not (comparing between OPT2 and OPT3), because $\bar{g}$ is only a function of the total available soil water (solid and dashed lines in Fig. 4A). This result occurs despite the fact that OPT2 and OPT3 are defined using different functional dependence of $g$ on $x$; i.e., the optimal stomatal conductance obtained from OPT3 (Eq. (13)) is higher in well-watered conditions, but decreases at low soil moisture (dashed line in Fig. 2C) compared to the model variant without soil moisture limitations (solid line in Fig. 2C). The $\bar{g}$ can be derived analytically by formulating the constraint that soil water is limited as a relation between total transpiration amount and available soil water,

$$\int_0^{t_d} \nu E(t)dt = w_0(x_0 - x_T). \tag{19}$$

Using the definition of temporal average, Eq. (19) can be written as,

$$\bar{E} = \frac{\int_0^{t_d} E(t)dt}{t_d} = \frac{w_0(x_0 - x_T)}{\nu t_d}. \tag{20}$$

Recalling Eq. (1) and (4), the mean stomatal conductance can thus be expressed as,

$$\bar{g} = \frac{\int_0^{t_d} g(t)dt}{t_d} = \frac{\int_0^{t_d} E(t)dt}{aDLt_d} = \frac{w_0(x_0 - x_T)}{\nu aDLt_d}, \tag{21}$$

which is independent of the specific trajectory $g(t)$, but it is indirectly dependent on $c_a$ via $L$.

Canopy-level net $CO_2$ assimilation rate increases with $c_a$ in all optimization models due to the direct $CO_2$ fertilization effect, but more so when leaf area acclimates (green *vs.* black lines in Fig. 4B), and at a higher rate with the instantaneous optimization approach (dot-dashed *vs.* solid lines in Fig. 4B). In contrast to the mean stomatal conductance, the mean net $CO_2$ assimilation rate does depend on whether soil water is limiting or not (i.e., the specific $g(t)$ matters), due to the nonlinear nature of the $A_L(g)$ relation (Eq. (3)). In particular, diminishing returns at high $g$ cause $\bar{A}$ to be lower when optimal $g$ from OPT3 is higher under well-watered conditions and lower in dry conditions, compared to OPT2 with time-invariant $g$. This explains why the dashed lines in Fig. 4B are lower than the corresponding solid lines.

Therefore, based on the results in Fig. 4, the inclusion of the dynamic feedback (OPT2 and OPT3) in the stomatal optimization model produces plausible responses to elevated $c_a$. The dynamic feedback variants are also more suitable given our focus on long-term responses of gas exchange. Conversely, the stomatal response to elevated $CO_2$ of OPT1 is not realistic because $\lambda$ is independent of $c_a$ (Fig. 4A; see also Sect. 4.4). In contrast, the responses of both dynamic feedback

approaches are plausible. In the following comparisons with the PETA model, we consider only the optimization model without any water limitation effect (OPT2), because the relative changes in gas exchange rates are essentially the same when including water limitation (OPT3; results not shown), despite variations in the absolute rates.

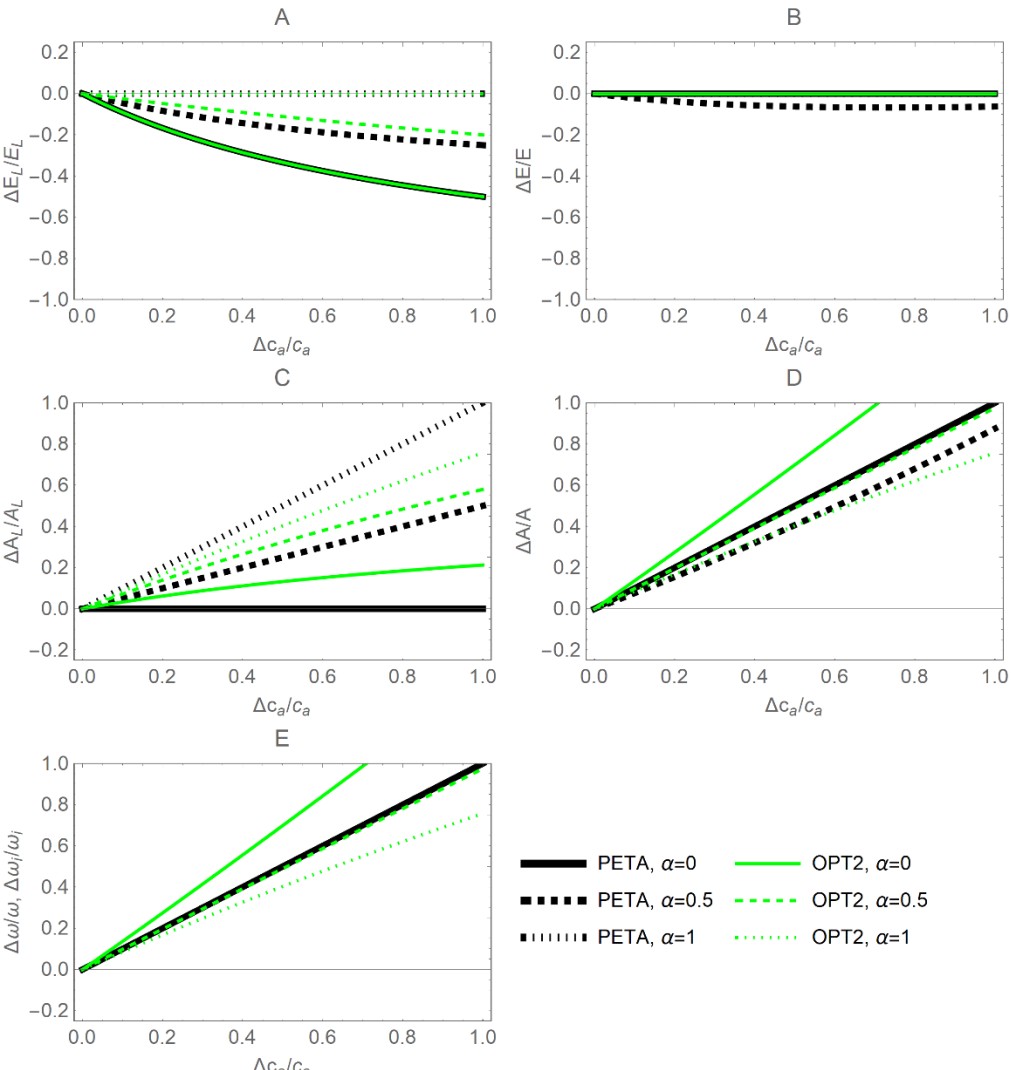


**Fig. 5. Relative changes in leaf-level (A, C) and canopy-level (B, D) gas exchange rates as a function of relative change in atmospheric $CO_2$ concentration $c_a$, as predicted by the PETA model (black lines) and the optimal stomatal control model OPT2 (green lines): A) leaf-level transpiration rate ($E_L$), B) canopy-level transpiration rate ($E$), C) leaf-level assimilation rate ($A_L$), and D) canopy-level assimilation rate ($A$), and E) water use efficiency ($\omega$, equivalent to intrinsic**
**WUE at constant VPD). Changes in $c_a$ have both direct and indirect effects on the $CO_2$ and water vapor exchange rates; the indirect effects are mediated by changes in leaf area that also depend on canopy status, indicated by $\alpha$ (Fig. 3): lower values of $\alpha$ refer to open-canopy conditions with largest leaf area stimulation by elevated $c_a$; for $\alpha = 1$ leaf area is constant. Vapor pressure deficit and dry period length are equal to the baseline values (Table 2).**

## 3.2 Gas exchange responses to changes in atmospheric $CO_2$ concentration

The relative variations of gas exchange rates and water use efficiency predicted under elevated $CO_2$ concentration by the PETA and optimization model with dynamic feedback but absence of water limitation (OPT2) are broadly consistent (Fig. 5). As $CO_2$ concentration increases, both models predict decreasing leaf-level (Fig. 5A, except for $\alpha = 0$), but stable canopy-level transpiration rates (Fig. 5B), and increasing net $CO_2$ assimilation rates at both leaf- and canopy-levels (Fig. 5C, D). Therefore, water use efficiency ($\omega$) increases with increasing atmospheric $CO_2$ concentration (Fig. 5E). In the PETA model,

the increase in $\omega$ is linear with $CO_2$ by definition (Eq. (5)), while it is slightly nonlinear for the optimization models. The predicted sensitivity of the gas exchange responses varies between PETA and optimization models, depending on the canopy status (i.e., $\alpha$), in particular for the rate of net $CO_2$ assimilation (Fig. 5C, D). At the leaf level, higher $\alpha$ reduces the sensitivity of transpiration rates, but enhances that of net $CO_2$ assimilation rates to increasing $CO_2$ concentration in both models (compare dotted and solid lines in Fig. 5A, C). In contrast, at the canopy-level, higher $\alpha$ reduces the net $CO_2$

assimilation responses to $CO_2$ concentration in the PETA model (Fig. 5D). Conversely, by construction, canopy-level transpiration is independent of atmospheric $CO_2$ according to the optimality model (Eq. (20); all green lines overlap on the $\Delta E/E = 0$ axis in Fig. 5B). By definition, $\omega$ is independent of $\alpha$ in the PETA model (all black lines are overlapping in Fig. 5E), whereas a more open canopy (lower $\alpha$) increases the sensitivity of $\omega$ to changes in $CO_2$ concentration according to the optimality model. In the following analyses, we prescribed the intermediate value $\alpha = 0.5$.

## 3.3 Gas exchange responses to combined changes in atmospheric $CO_2$ concentration, VPD, and dry period length


The gas exchange patterns driven by $c_a$ and $D$ are largely consistent between the PETA and optimization models. In both PETA and OPT2 models, at a given $c_a$, higher VPD slightly increases leaf-level transpiration (Fig. 6A, F, K), but has no effect on canopy-level transpiration (Fig. 6B, G, L) because leaf area decreases with increasing VPD (Eq. (5) and (6)). The decrease in stomatal conductance at higher VPD in both models, and irrespective of how the change in VPD is imposed,

causes the intrinsic water use efficiency to increase (Fig. 6E, J, O). Moreover, higher VPD decreases leaf- and canopy-level net $CO_2$ assimilation when VPD is varied at fixed temperature (Fig. 6C-D for PETA, H-I for OPT2). However, when VPD is varied because of changing temperature (which also affects photosynthetic parameters; bottom row), leaf-level net $CO_2$ assimilation increases and then decreases slightly as VPD is increased, whereas canopy-level net $CO_2$ assimilation decreases (Fig. 6M, N). Following a hypothetical climate change trajectory with simultaneous increases in $c_a$ and $D$ (arrows in Fig. 6),

higher VPD reduces the improvement in canopy-level net $CO_2$ assimilation rate caused by elevated $CO_2$ alone, while leading to a greater improvement in intrinsic water use efficiency.

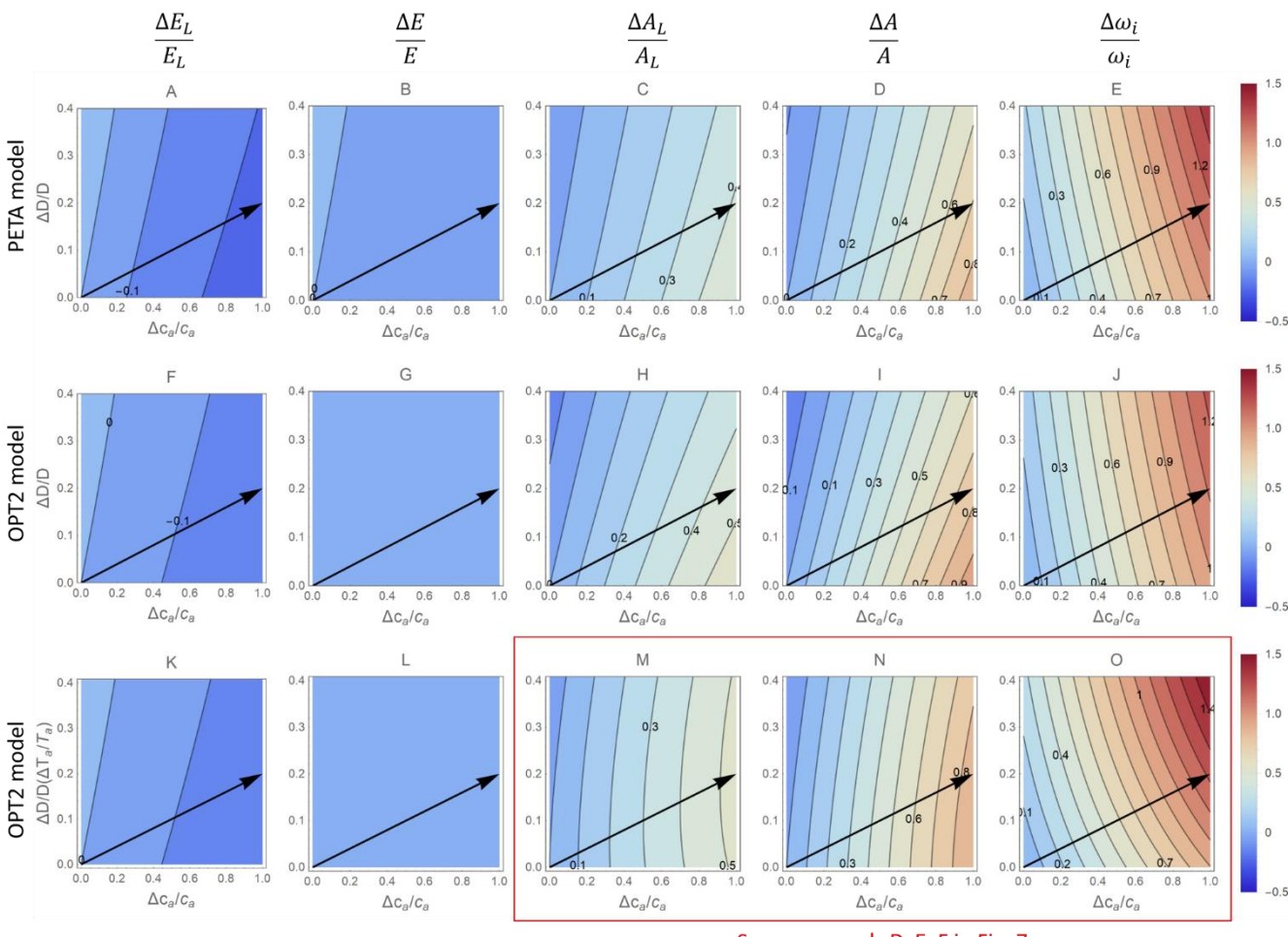

**Fig. 6.** Contour plots of relative changes in leaf-level (A, C, F, H, K, M) and canopy-level (B, D, G, I, L, N) gas exchange rates as a function of relative changes in atmospheric $CO_2$ concentration $c_a$ (x-axis) and vapor pressure deficit $D$ (y-axis), as predicted by the PETA model (top panels) and the optimal stomatal control model OPT2 (centre and bottom panels): A, F, K) leaf-level transpiration rate ($E_L$); B, G, L) canopy-level transpiration rate ($E$); C, H, M) leaf-level assimilation rate ($A_L$); D, I, N) canopy-level assimilation rate ($A$); and E, J, O) intrinsic water use efficiency ($\omega_i$). In F-J, $D$ is varied by letting the relative humidity change at constant temperature $T_a$ (i.e., the assimilation rate constants do not co-vary with $D$); in K-O, $D$ is varied by letting the $T_a$ change at constant relative humidity, set at 50% (i.e., the assimilation rate constants co-vary with $D$ due to the effect of $T_a$). Leaf area index varies with $c_a$ and $D$ according to Eq. (6) with $\alpha =0.5$. Black arrows indicate hypothetical temporal trends in $D$ and $c_a$ assuming a $CO_2$ concentration doubling and associated $D$ and $T_a$ increase. The dry period length is assumed to be constant and equal to the baseline value (Table 2).

While the responses of transpiration rates are the same regardless of how the variation in VPD is produced, patterns in net $CO_2$ assimilation rates (and thus also water use efficiency) depend strongly on the selected baseline temperature in the optimization model, as shown in Fig. 7. Here, only results from the optimization model OPT2 are shown, because the PETA model cannot attribute variations in VPD to relative humidity or temperature. At low baseline $T_a$ (top row), higher VPD enhances net $CO_2$ assimilation because changes in VPD are driven by temperature increases that also promote photosynthesis (i.e., the baseline $T_a$ is below the photosynthetic thermal optimum). In contrast, at high baseline $T_a$ (bottom row), temperature increases driving VPD inhibit photosynthesis (i.e., the baseline $T_a$ is above the photosynthetic thermal optimum). The case shown in the central row (same as in Fig. 6) is intermediate between these two extremes. As a result, simultaneously increasing VPD and $c_a$ along the arrows in Fig. 7 causes a faster or slower increase in net $CO_2$ assimilation than would occur due to changes in $c_a$ alone, depending on whether the baseline temperature is lower or higher than the thermal optimum, respectively. Accordingly, with increasing baseline $T_a$, the $c_a$-driven enhancement of intrinsic water use efficiency also decreases (Fig. 7C, F, I).

Changing the length of the mean dry period leads to contrasting responses of the PETA and optimization models (Fig. 8), mostly because PETA does not include any effect of soil moisture on the $CO_2$ responses (i.e., predicted responses are independent of $t_d$; Fig. 8A-E). In the optimization model, for a given $c_a$, longer dry periods lower all gas exchange rates (Fig. 8F-I), while increasing the intrinsic water use efficiency (Fig. 8J). Following a hypothetical trajectory of increasing $c_a$ and $t_d$ (solid arrows in Fig. 8F-J), the lengthening of the dry periods—similar to increasing VPD—reduces the positive effect of elevated $CO_2$ on net $CO_2$ assimilation compared to a scenario where only $c_a$ is increased. The opposite pattern occurs if we assume wetting (shorter $t_d$) is associated with elevated $CO_2$ (dashed arrows in Fig. 8F-J).

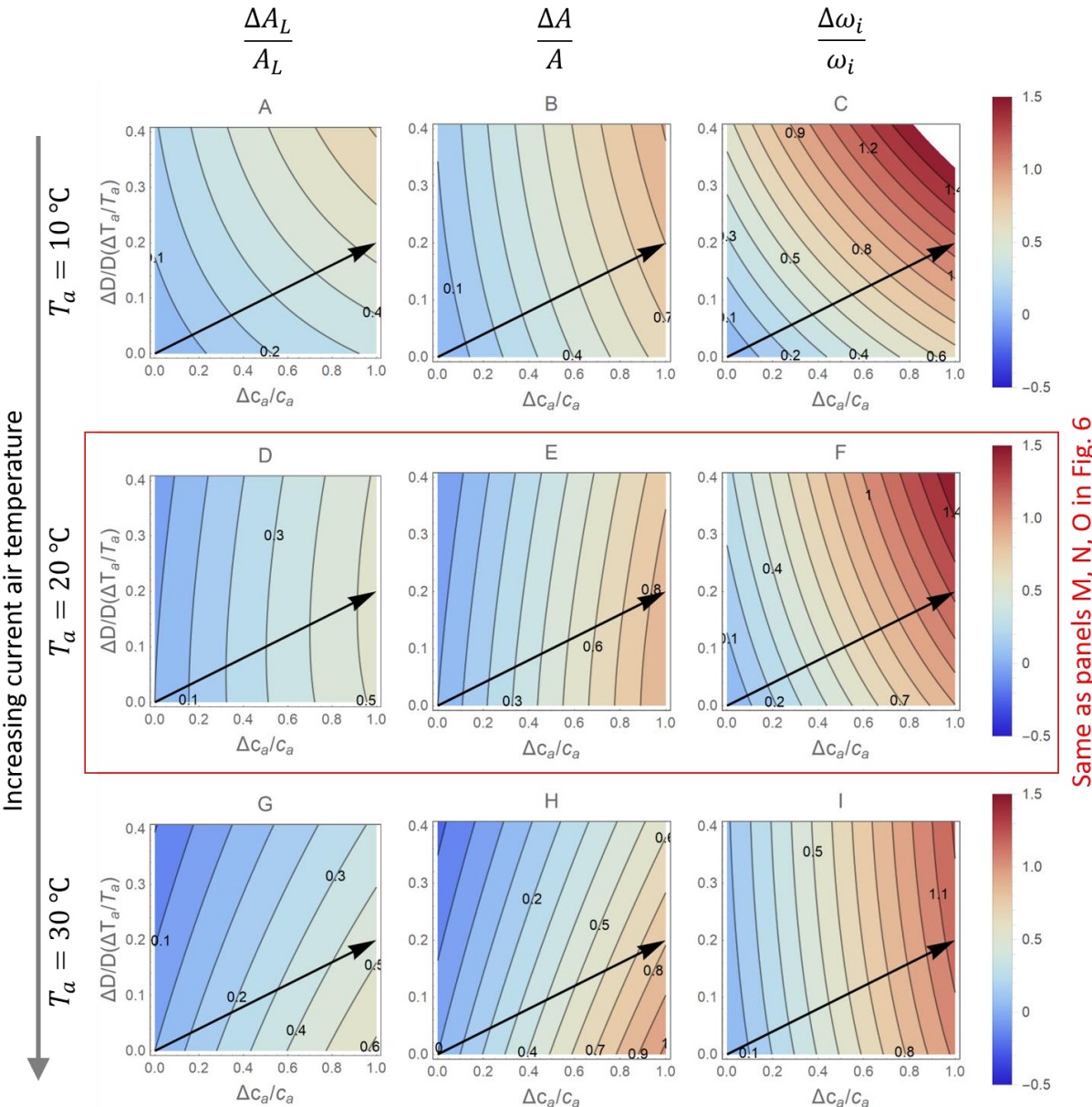


**Fig. 7. Contour plots of relative changes in leaf- ($A_L$; A, D, G) and canopy-level ($A$; B, E, H) net $CO_2$ assimilation rates, as well as intrinsic water use efficiency ($\omega$; C, F, I) as a function of relative changes in atmospheric $CO_2$ concentration $c_a$ (x-axis), and vapor pressure deficit $D$ (y-axis), as predicted by the optimal stomatal control model OPT2. The baseline temperature used to calculate relative changes is increased from top ($T_a$=10 °C) to bottom (30 °C), with the central panels corresponding to panels M, N and O in Fig. 6 ($T_a$=20 °C). Changes in VPD are driven by temperature $T_a$ at constant relative humidity (increasing from top to bottom to keep the same baseline VPD). Other parameters are as in Fig. 6.**

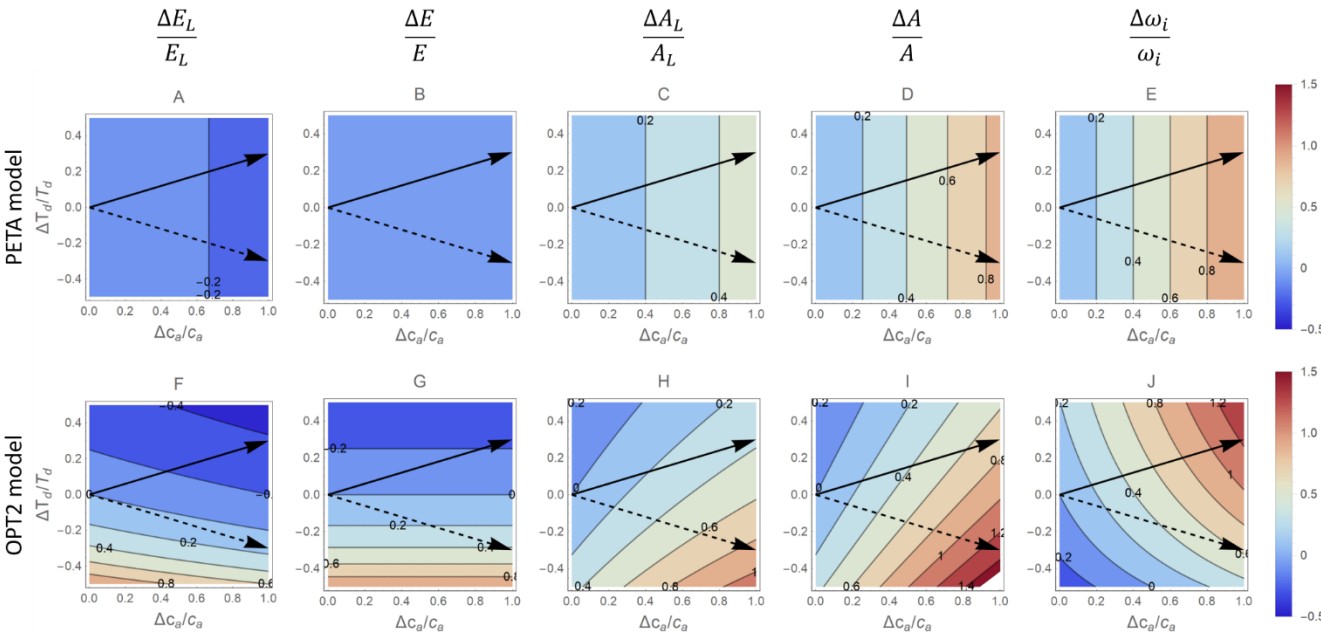


**Fig. 8. Contour plots of relative changes in leaf-level (A, C, F, H) and canopy-level (B, D, G, I) gas exchange rates as a function of relative chances in atmospheric $CO_2$ concentration $c_a$ (x-axis) and dry period length $t_d$ (y-axis), as predicted by the PETA model (top panels) and the optimal stomatal control model OPT2 (bottom panels): A, F) leaf-level transpiration rate ($E_L$); B, G) canopy-level transpiration rate ($E$); C, H) leaf-level assimilation rate ($A_L$); D, I) canopy-level assimilation rate ($A$); and E, J) intrinsic water use efficiency ($\omega_i$). Leaf area index varies with $c_a$ and $D$ according to Eq. (6) with $\alpha = 0.5$. Black arrows indicate hypothetical temporal trends in $t_d$ and $c_a$ in locations where $t_d$ will lengthen (solid arrow) or shorten (dashed arrow) as $c_a$ increases. The vapor pressure deficit is assumed to be constant and equal to the baseline value (Table 2).**

## 4    Discussion

### 4.1  Water availability constrains leaf and canopy transpiration responses to atmospheric $CO_2$ (question 1)

Vegetation acclimates and adapts to increasing atmospheric $CO_2$ concentration by adjusting tissue-level traits, biomass allocation and, ultimately, community composition. Even in a $CO_2$-fertilized world, several other resources might limit vegetation growth, including light, nutrients, and water. It is therefore reasonable to expect that growth patterns will adjust so that the available resources are used effectively. These adjustments might occur at different biological levels and temporal scales (organ, whole plant, community) and can be large and possibly of opposite sign. However, we can expect that their net effects converge towards an effective use of any limiting resource in addition to carbon. As a result, despite potentially large variations in individual plant traits, limiting resources would be utilized to the maximum extent possible. In other words, quoting out of context, "Se vogliamo che tutto rimanga com'è bisogna che tutto cambi." [For everything to remain as it is, everything must change.] (di Lampedusa G. T., 1958, "Il Gattopardo").

Both PETA and dynamic feedback optimization models predict that in fully acclimated plants and for a given soil water availability and VPD, increasing atmospheric $CO_2$ concentration will cause a decrease in leaf-level transpiration and have no effect on transpiration at the canopy level. This is in contrast to short-term responses in which stomatal conductance and thus leaf-level transpiration were observed to decrease under elevated $CO_2$ concentrations, when plants are not yet fully acclimated. However, PETA and optimization model predictions are consistent with both long-term observations in

presumably fully acclimated plants (Schäfer et al., 2002) and results from other, more detailed models (Fatichi et al., 2016). The decreased sensitivity of transpiration rate to elevated $CO_2$ is expected in the long-term when allowing plant or community-averaged traits besides stomatal conductance to optimally acclimate (or adapt), because constraints in resources other than $CO_2$ become important and ultimately determine gas exchange and plant growth (Schymanski et al., 2015). Predicting long-term gas exchange under elevated $CO_2$ thus requires considering the full spectrum of plant adjustments,

especially in ecosystems where water is a known limiting factor.

If indeed plants adjust leaf area and stomatal conductance to use the available water, in semiarid or seasonally dry ecosystems, soil moisture values should be stable in long-term $CO_2$ enrichment experiments. However, soil moisture can be higher under elevated $CO_2$ conditions, contradicting the assumption of the optimization model (Lu et al., 2016a; Fay et al., 2012). This increased water availability might occur only in the short-term because $CO_2$ enrichment had not been running

long enough for plants and communities to fully acclimate. Moreover, our simplified model does not include intra- or inter-specific competition occurring in response to elevated $CO_2$ (e.g., Fay et al., 2012), which can alter water use strategies by intensifying water consumption at high soil moisture (Manzoni et al., 2013), and therefore cause a deviation from the optimal stomatal conductance behaviour we derived here. Other empirical evidence instead support the assumption that soil water is a main constraint for transpiration—especially in water-limited ecosystems where atmospheric demand is high and

where evapotranspiration tends to match precipitation on an annual basis (Williams et al., 2012), or even exceed it during the growing season due to soil water storage.

Both PETA and optimization models predict increasing leaf- and canopy-level net $CO_2$ assimilation rates with increasing $c_a$—a well-known response (Ainsworth and Long, 2005; Norby et al., 1999). As a consequence of combined changes in transpiration and net $CO_2$ assimilation, WUE and iWUE also increase. Indeed, changes in WUE estimated from flux-towers

and isotope composition of tree rings can be more than proportional (Keenan et al., 2013; Mastrotheodoros et al., 2017) or almost proportional to changes in $c_a$ (Dekker et al., 2016; Frank et al., 2015; Lavergne et al., 2019). Our results suggest relative changes in iWUE between 0.15 and 0.29 % ppm$^{-1}$ with the lower values when VPD is assumed fixed and higher values when it increases together with $CO_2$ concentration (Fig. 6 and 7). Values reported in previous studies tend to overlap to this range or be higher: 0.22-0.35 % ppm$^{-1}$ (for broadleaf and conifers, respectively, Frank et al., 2015), 0.3-0.75 % ppm$^{-1}$

(with variation between angiosperms and conifers, and among climates, Adams et al., 2020), 0.41 % ppm$^{-1}$ (Penuelas et al., 2011), 0.44 % ppm$^{-1}$ (Saurer et al., 2014), 0.52 % ppm$^{-1}$ (Dekker et al., 2016). Our estimates were obtained without any parameter adjustment (for the PETA model, only $\alpha$ could be adjusted; for the optimization model, physiological and soil

parameters could be varied within reasonable ranges). Therefore, we consider the predictions of iWUE sensitivity accurate, given the simplicity of our approach.

## 560    **4.2 Atmospheric CO₂ and vapor pressure deficit interact in defining gas exchange responses (question 2)**

The effect of elevated atmospheric $CO_2$ is mediated by changes in other environmental variables related to water availability, such as VPD and the duration of dry periods. For a given $c_a$, increasing VPD has little or no effect on transpiration rates because, in the PETA model, relative changes in VPD have small effects on WUE (they appear under the square root of Eq. (5)), and hence on $E_L$ (Eq. (6)). If gas exchanges were only controlled by diffusion (without leaf internal $CO_2$ drawdown by photosynthesis), VPD would have a stronger effect on transpiration rates, as shown in Appendix A for the case of PETA model. Similarly, minor VPD effects in the optimization model are due to soil water constraining transpiration, with stomatal conductance adjusting accordingly. Indeed, because of this constraint, $g \sim D^{-1}$, where $D$ is interpreted as the long-term mean VPD (Eq. (21)). Had we calculated $\lambda$ from long-term environmental conditions (so that $\lambda$ is a constant in OPT2 or OPT3), and then let VPD vary for given $c_a$, LAI and other conditions, to simulate short-term VPD responses, we would have instead obtained $g \sim D^{-1/2}$, consistent with observations in short-term measurements. In fact, the declines of stomatal and canopy conductance with increasing $D$ when all other environmental conditions are fixed was well-captured by $g \sim 1 - m \log(D)$ with $m = 0.5\text{-}0.6$ (Oren et al., 1999). This logarithmic relation can be approximated by $g \sim D^{-1/2}$ (Katul et al., 2009). Confirming these results, in a recent meta-analysis, increasing VPD decreased $g$ and net $CO_2$ assimilation rate, but increased leaf transpiration rate (Lopez et al., 2021). However, in the same study, plant-level transpiration rate also increased with VPD, with a saturating effect, which is in contrast with the model-predicted small increase (according to PETA) or no change (according to optimization) of $E$ as VPD increases (Fig. 6). More complex canopies and structural adjustments not considered here—e.g., rooting depth (see Appendix C)—might allow plants to access more water when the evaporative demand is higher, explaining higher than predicted plant-level transpiration in that meta-analysis.

Reductions in $g$ cause less than proportional reductions in net $CO_2$ assimilation rates (Eq. (3)), resulting in increasing iWUE with increasing VPD for a given $c_a$. Such a response was observed at the ecosystem level, regardless of changes in soil moisture, leading to the projection (under RCP 8.5) that iWUE could increase by 10% to 35% by 2100 because of the increase in VPD alone (Zhang et al., 2019), in line with results in Fig. 6.

Increasing VPD (driven by either temperature or relative humidity) in conjunction with $c_a$ has limited effects on transpiration rates and increases the sensitivity of iWUE to $c_a$ in both models (Fig. 6), whereas the sensitivity of net $CO_2$ assimilation varies with temperature in the optimization model (Fig. 7). This temperature effect is caused by the direct temperature dependence of photosynthetic kinetics (Medlyn et al., 2002) and the indirect effect via VPD. As the growth temperature is increased (i.e., moving towards lower latitudes), the optimization model predicts decreasing sensitivity of net $CO_2$ assimilation to changes in $c_a$ when VPD variations are driven by warming. Lower sensitivities at high growth temperatures are due to negative effects of warming on photosynthesis implemented in the model, as the growth temperature moves

beyond the thermal optimum of photosynthesis. Accounting for thermal acclimation and different thermal optima depending on growth conditions (Vico et al., 2019; Smith et al., 2020) could compensate for this decline in sensitivity, but warming could also have other consequences that are not considered here. For example, warming can lengthen the growing season, and change nutrient availability and biomass allocation to leaves *vs.* roots (Way and Oren, 2010), which in turn might affect the equilibrium LAI and photosynthetic capacity. Considering all these factors is beyond the scope here, where we restricted

temperature effects to the kinetics of photosynthesis and warming-induced air drying.

## 4.3 Atmospheric $CO_2$ and dry-down duration interact in defining gas exchange responses (question 2)

The dry-down duration affects the gas exchange response to elevated $c_a$ only in the optimization model OPT2, where $t_d$ appears explicitly in the equations. Not surprisingly, longer dry periods cause stomatal conductance to be downregulated, resulting in decreased gas exchange rates, while shorter ones increase them. This result is perhaps best understood by

considering Eq. (21), where all else being equal, $\bar{g} \sim t_d^{-1}$. This prediction is a consequence of the assumption that plants have evolved to use all soil water during the hypothetical dry-down of duration $t_d$, and that the total water storage during the dry period is fixed regardless of its duration. If longer $t_d$ were instead associated with incomplete recharge resulting in lowered initial soil moisture $x_0$, the exponent of the $\bar{g}$ *vs.* $t_d$ relation would be even more negative. As a result, all gas exchange rates would decrease with lengthening of $t_d$ faster than in Fig. 8. Notably, longer dry periods increase WUE because as stomata

close, the slope of the $A_L(g)$ relation in our simple model steepens (Eq. (3)). In fact, Eq. (2) suggests that for $g/k \gg 1$, $A_L(g) \approx k\, c_a$, $\partial A_L/\partial g \approx 0$ (a minimum slope corresponding to no stomatal limitation). Conversely, when $0 < g/k \ll 1$, $A_L(g) \approx g\, c_a$, $\partial A_L/\partial g \approx c_a$, which is the maximum attainable slope when all $CO_2$ taken up is also assimilated.

While typical rain exclusion experiments alter rewetting intensities more than dry period durations, rainfall manipulations where the same amount of water is concentrated into fewer, more intense events could provide a suitable testing ground for

these predictions. The advantage of these experiments compared to observations along a natural climatic gradient is that all conditions except rainfall timing and amount are the same, as in our numerical experiments where we let one or two factors vary at a time. Consistent with model results, both net $CO_2$ assimilation rates and stomatal conductance decrease when rainfall frequency is reduced in a grassland ecosystem (Knapp et al., 2002; Fay et al., 2002). These reduced gas exchanges lower plant productivity, but also promote allocation to roots when rainfall frequency is reduced (Fay et al., 2003),

suggesting that flexible allocation to belowground tissues might complement the stomatal conductance and leaf area adjustments that are the focus of the simple models used here. Lower rainfall frequency (for given total precipitation) can also increase productivity in semi-arid ecosystems where fewer larger events promote soil moisture thanks to higher infiltration and lower evaporation from the soil surface (Heisler-White et al., 2008). These factors in the water balance were not explicitly considered here, but can be important to determine the amount of available water, which in turn is the key

constraint for stomatal responses to elevated atmospheric $CO_2$.

## 4.4  Model assumptions and limitations

The choice of the specific limiting factor for photosynthesis leads to a range of optimal stomatal conductance solutions as a function of the Lagrange multiplier $\lambda$ and other environmental conditions. Equation (3) assumes that the net $CO_2$ assimilation rate depends linearly on leaf internal $CO_2$ concentration, with an additional effect of atmospheric $CO_2$ concentration that allows capturing the nonlinear nature of the $A$-$c_i$ curve. Other assumptions can be imposed, including light-limited (Medlyn et al., 2011) or $CO_2$ and light co-limited photosynthesis (Vico et al., 2013; Dewar et al., 2018). The resulting stomatal conductance can be mathematically similar to or different from Eq. (10), and in particular with contrasting dependencies on atmospheric $CO_2$ concentration. For example, the optimization model OPT2 that we selected for its mathematical simplicity does not correctly predict the short-term stomatal closure observed when atmospheric $CO_2$ concentration is increased (Fig. 4A). This is a known pathology of this formulation (Medlyn et al., 2011; Katul et al., 2010; Buckley and Schymanski, 2014), but assuming RuBP-limited photosynthesis or co-limitation also leads to the same issue, even though it appears at lower $c_a$ (Vico et al., 2013; Dewar et al., 2018). Interestingly, also optimizing $c_i/c_a$ to maximize carbon gains minus water transport costs per unit of net $CO_2$ assimilation (Prentice et al., 2014) results in increasing stomatal conductance with $c_a$ at pre-industrial $c_a$ values (Fig. S2 in Joshi et al., 2022). In the stomatal optimization models, these erroneous responses arise because at low $CO_2$ concentration a small increase in stomatal conductance results in large net $CO_2$ assimilation gains compared to the higher water losses, resulting in the counterintuitive opening of stomata as atmospheric $CO_2$ concentration is increased. This issue appears when $\lambda$ is fixed (i.e., using the instantaneous optimization approach without acclimation), instead of being determined while solving the optimization problem or being heuristically increased at higher $CO_2$ concentration (Katul et al., 2010; Manzoni et al., 2011).

As long as the Hamiltonian of the optimization problem is independent of soil moisture (i.e., $\partial(A - \lambda E)/\partial x = 0$), the Lagrange multiplier is time invariant ($d\lambda/dt = 0$) because a necessary condition for the optimization is $d\lambda/dt = -\partial(A - \lambda E)/\partial x$ (Manzoni et al., 2013). The numerical value of this time invariant $\lambda$ can be determined by imposing the condition that all available water is used by the end of the dry period. Accounting for this constraint and thus calculating $\lambda$ in Eq. (10) (or any analogous formulations based on other assumptions) leads to an optimal stomatal conductance value that essentially reflects the constraint imposed on water availability (Eq. (11) or (13))—regardless of the assumed kinetics of photosynthesis. In turn, this means that any assumption on the factor limiting photosynthesis will lead to the same optimal stomatal conductance value as long as the Lagrange multiplier is solved for within the optimization problem. Therefore, the predictions of the optimization model after imposing the constraint of limited water availability are expected to be similar for any choice of the net $CO_2$ assimilation model.

Other models based on instantaneous maximization of C gains for given costs offer alternative frameworks to predict responses to atmospheric $CO_2$ concentrations and other environmental changes (Sperry et al., 2017; Mencuccini et al., 2019; Huang et al., 2018; Bassiouni and Vico, 2021; Prentice et al., 2014; Joshi et al., 2022). For example, the model based on Prentice et al. (2014) correctly predicts the observed short-term decrease in stomatal conductance under elevated (but not

pre-industrial) atmospheric $CO_2$ (Eq. (C1) in Stocker et al., 2020) without invoking leaf area adjustments. While these approaches are physiologically plausible in the way they balance instantaneous C gains and losses and their predictions compare well with observed trends, they do not guarantee that the water use is optimal over a given time interval. In other words, instantaneous maximization models rest on the assumption that future C gains are so uncertain that maximizing short-term gains is more convenient (in an evolutionary sense). In contrast, models based on optimal control theory rest on the assumption that future gains are expected because climatic conditions are to some degree predictable (rain on average occurs every $t_d$ days), or that plant responses have been adapted to 'anticipate' these long-term, probabilistic conditions. These approaches can be seen as end-member cases along a continuum or possible optimization strategies.

In more complex models, it was assumed that not only stomatal conductance, but also LAI or rooting depth were optimized to reach a certain objective (typically maximize long-term productivity) (Schymanski et al., 2015). Here instead, LAI was prescribed—not optimized—as a function of $c_a$ and environmental conditions as reflected by $\alpha$. Combining stomatal and leaf area optimization would have resulted in a more complex model that would have been difficult to compare to the PETA model. Rooting depth or root density were also not optimized, nor were they varied in the analyses shown in Fig. 5-8, as they are not included as parameters in the PETA model. However, deeper or more dense roots might allow access to a larger soil water store. If elevated $CO_2$ increases leaf area and plant size overall, allometric relations would predict a corresponding increase in root biomass and spatial extent (see Chapter 6 in Hunt and Manzoni, 2015; Kempes et al., 2011). Consistent with this expectation, an optimality model predicted deeper roots and higher root area indices under elevated $CO_2$, which supplied water to support higher transpiration rates than seen under ambient $CO_2$ (Schymanski et al., 2015). These arguments are developed in Appendix C, where we show that the optimal stomatal conductance would be less sensitive to elevated $CO_2$ compared to Fig. 5-8 if deeper roots develop under elevated $CO_2$, resulting in a slight positive effect elevated $CO_2$ on transpiration. However, these deviations are minor for realistic values of the exponent of the rooting depth *vs.* leaf area index relation.

Besides root allocation, we also neglected evaporation from the soil or canopy surface. Changes in LAI do not affect strongly the partitioning of evapotranspiration into transpiration and evaporation, thanks to two compensating mechanisms—with increasing LAI, interception and subsequent evaporation from leaf surfaces increase, while heating of the soil surface is reduced, thus also reducing evaporation (Fatichi and Pappas, 2017; Paschalis et al., 2018). Therefore, even without explicitly modelling evaporation from the soil, the relative changes in gas exchange (as presented here) should be correctly predicted.

For simplicity, we restricted our analysis to deterministic conditions—a single 'representative' dry-down with prescribed initial and final soil moisture states, and duration. Clearly, all these features of dry periods should be treated as stochastic, because rainfall timing and amounts are inherently stochastic (Rodriguez-Iturbe and Porporato, 2004). Stomatal optimization can be studied also in a stochastic rainfall scenario consisting of consecutive dry-downs of random initial states and durations, where rainfall is characterized by a constant mean event frequency and daily intensity. Under long-term steady state conditions, the optimization of $CO_2$ assimilation integrated over an infinite time period can be replaced by the integral over all possible states of the stochastic processes (i.e., over all values of stochastic soil moisture) (Lu et al., 2016b, 2020).

The resulting solution reflects the expected stomatal behaviour under the probabilistic (in contrast to deterministic) temporal evolution of soil moisture. Stomatal conductance and transpiration rate were predicted to increase with mean annual precipitation (especially so with high rainfall frequency for given total precipitation), with a saturation effect at high precipitation. Moreover—and consistent with our results—optimal water use under stochastic rainfall was not predicted to change under elevated atmospheric $CO_2$. Similarly, plants should evolve towards more intensive use of water when rainfall frequency or amount per event increase, at lease in recruitment limited plant communities (Lindh and Manzoni, 2021). This effect is qualitatively similar to our prediction of higher transpiration with increasing water storage capacity.

## 5 Conclusions

Despite increasing atmospheric $CO_2$ concentration and VPD, only small changes in canopy-scale evapotranspiration have been observed or predicted (Fatichi et al., 2016; Knauer et al., 2017; Yang et al., 2021). That long-term transpiration is a 'conserved' hydrological quantity had been already noted when comparing forests under current climatic conditions (Roberts, 1983), suggesting that vegetation acclimates in such a way as to maintain stable transpiration under a given climate. This behaviour could be the result of a number of compensatory feedback mechanisms, including acclimation of leaf area together with stomatal conductance. We quantified the consequences of simultaneous changes in stomatal conductance and leaf area on gas exchange by means of two analytical models of stomatal conductance: PETA and stomatal optimization. Both models predict low sensitivity of canopy transpiration rates to a changing climate, indicating that morphological adjustments (leaf area increase) compensate physiological adjustments (stomatal closure). However, this similar outcome is due to different reasons. In the PETA model, this was the result of a set of heuristic assumptions on how gas exchange varies with leaf area and water use efficiency, whereas, in the optimization model, this stemmed from water availability setting constraints on canopy transpiration. Moreover, when leaf area increases in response to elevated $CO_2$, stomata close according to the optimization model, regardless of the chosen formulation for net $CO_2$ assimilation. With stable transpiration and predicted increases in net $CO_2$ assimilation rates in both models, intrinsic water use efficiency is also predicted to increase under elevated $CO_2$. Finally, the sensitivity of net $CO_2$ assimilation, and to some degree of intrinsic water use efficiency, to changes in $CO_2$ concentration are mediated by warming-induced increases in VPD. Drier air is expected to enhance the positive effect of elevated $CO_2$ concentrations on net $CO_2$ assimilation and water use efficiency, but only at growth temperatures lower than the photosynthetic thermal optimum, whereas the effect of rising $CO_2$ concentration is reduced at higher growth temperatures. Increases of VPD, air temperature and dry-down durations may have all contributed to the observation that the rate of intrinsic water use efficiency has increased more than proportionally to the current rise in atmospheric $CO_2$ levels. Overall, these results imply that physiological and morphological traits acclimate to changing environmental conditions in a coordinated manner to ensure that limiting resources such as water are used efficiently.

## Appendix A: Separating diffusion and biochemical limitations to net assimilation using a simplified PETA model

To support the arguments in Section 4.2, a simplified version of the PETA model is derived here considering that, in free air $CO_2$ enrichment experiments, $\chi = c_i/c_a$ is roughly constant at a fixed VPD (Ainsworth and Long, 2005). This leads to $\omega \sim c_a/D$ instead of $\omega \sim c_a/\sqrt{D}$ as postulated above to derive Eq. (5). This simplification is equivalent to ignoring the dependence of the intercellular to ambient $CO_2$ concentration ratio on $D$ (i.e., $1 - \chi$ is constant), and attribute the sensitivity to $D$ to only diffusion through the stomata. With this assumption, a simplified PETA model is obtained in which,

$$\frac{\Delta\omega}{\omega} = \frac{1+\frac{\Delta c_a}{c_a}}{1+\frac{\Delta D}{D}} - 1. \tag{22}$$

This simplified model can be used to separate the effects of diffusion limitations to gas exchange from either diffusion and biochemical limitations (using the full PETA model with $\omega$ calculated from Eq. (5); Fig. 6). By promoting $CO_2$ transport from the atmosphere to the leaf, biochemical demand lowers the negative effect of stomatal closure at high VPD. Therefore, the combined effects of stomatal closure and biochemical limitations, which draw down leaf internal $CO_2$ concentrations, would reduce the sensitivity of net $CO_2$ assimilation and leaf and canopy transpiration to higher VPD at a fixed $c_a$. In fact, combining the simplified Eq. (22) with Eq. (6), we find $\Delta E_L/E_L \sim (1 + \Delta D/D)(1 + \Delta c_a/c_a)^{-1}$, suggesting a stronger increase in $E_L$ with VPD compared to the case of compound diffusion and biochemical demand (i.e., $\Delta E_L/E_L \sim (1 + \Delta D/D)^{1/2}(1 + \Delta c_a/c_a)^{-1}$). The relative change in leaf net assimilation ($\Delta A_L/A_L \sim \Delta\omega/\omega$, Eq. (6)) scales as $(1 + \Delta c_a/c_a)(1 + \Delta D/D)^{-1/2}$ when biochemical demand is accounted for (Eq. (5)) and as $(1 + \Delta c_a/c_a)(1 + \Delta D/D)^{-1}$ when it is not (Eq. (22)). Taking the ratio, we find that biochemical demand changes $\Delta A_L/A_L$ by a factor $(1 + \Delta D/D)^{-1/2}$ and $\Delta E_L/E_L$ by a factor of $(1 + \Delta D/D)^{1/2}$ compared to the case of simple gas diffusion, indicating that biochemical demand increases the sensitivities of gas exchange when increasing VPD.

## Appendix B: Derivation of the stomatal optimization models

To set up the optimal stomatal conductance model, we start from the assumption that plants regulate stomatal conductance ($g$) to maximize canopy-level net assimilation ($A$) during a typical dry-down period ($t_d$),

$$J = \int_0^{t_d} A(g(t), x(t), t)dt. \tag{23}$$

Because soil moisture ($x$) is depleted as plants transpire, the soil water balance (Eq. **Error! Reference source not found.**) is included as a constraint to the optimization. Maximizing $CO_2$ assimilation at the leaf level would be mathematically equivalent (see Eq. (4)), since leaf area index is not treated as a control variable, but as a time-invariant parameter during a

dry-down (as in e.g., Manzoni et al., 2013). However, plants can still alter allocation and thus leaf area in response to atmospheric $CO_2$ concentration at climatic time scales (years to decades), which are much longer than the daily to weekly scales at which the optimization problem is formulated. This means that changes in leaf area are treated as a change the model parameter $L$. In Eq. (23), the leaf net $CO_2$ assimilation rate is explicitly written as a function of $g$ and $x$ to emphasize the dependence of both on the control variable ($g$) and the state variable representing the constraint ($x$). This optimal control

problem can be solved by using the Euler-Lagrange formulation that reduces to maximizing the Hamiltonian ($H$) with respect to $g$. That is, defining the Hamiltonian as $H = A + \lambda(-E)$, we obtain,

$$\frac{d}{dt}\left(\frac{\partial H}{\partial \dot{g}}\right) - \frac{\partial H}{\partial g} = 0 \Rightarrow \frac{\partial H}{\partial g} = 0 = \frac{\partial A}{\partial g} - \lambda \frac{\partial E}{\partial g}, \tag{24}$$

where the first term on the left-hand side of Eq. (24) is ignored because $H$ is independent of $\dot{g} = \partial g/\partial t$; $\lambda$ is the Lagrange multiplier, and in the second term $E$ is the sum of all fluxes of water lost from the soil (in this case, only the transpiration rate), expressed in mol $H_2O$ (m$^2$ ground)$^{-1}$ s$^{-1}$. With this choice of units for the water loss term, $\lambda$ is expressed in µmol $CO_2$

(mol $H_2O$)$^{-1}$. Other choices for the units of $A$ and $E$ would not affect the results of the following calculations, except for the numerical value of $\lambda$. Three variants of the optimization model can now be described, as explained in Section 2.3 and illustrated in Fig. 1: i) instantaneous optimization (undetermined $\lambda$; OPT1), ii) dynamic feedback optimization with transpiration continuing till plant-available soil water is depleted ($\lambda$ derived mathematically; OPT2), and iii) dynamic feedback with transpiration reduced in dry soil ($\lambda$ derived mathematically; OPT3). In this appendix we focus on the

derivations of OPT2 and OPT3.

### B1. Derivation of OPT2: dynamic feedback optimization with transpiration rate independent of soil moisture

A more realistic approach that overcomes the limitation of a freely adjustable $\lambda$ is determining the value of $\lambda$ by imposing the constraint that the initial soil moisture $x_0$ is depleted, leaving only $x_T$ at the end of the time interval $t_d$. This means that we impose $x(t = t_d) = x_T$ as the soil moisture at the end of the dry-down described by Eq. **Error! Reference source not**

**found.**, where transpiration depends on $g_{opt}(\lambda)$ from Eq. (10); i.e., $\int_0^{t_d} vE(t)dt = \int_0^{t_d} va g_{opt}(\lambda)DLdt = w_0(x_0 - x_T)$. With this constraint in place, the only unknown is $\lambda$, which is found as (Manzoni et al., 2013),

$$\lambda = c_a a D \left[ \frac{w_0(x_0 - x_T)}{kLt_d v} + aD \right]^{-2}. \tag{25}$$

The linear scaling of $\lambda$ with $c_a$ in Eq. (25) is not externally imposed (as in Katul et al., 2010), but is an emergent property of the optimization with limited water availability. In this sense, $\lambda$ is not simply an adjustable parameter (as it has been treated previously, as in OPT1), but rather a clearly defined property of the coupled soil-plant system, including the amount of water

available in the soil. Substituting Eq. (25) into Eq. (10) and (3), the values of optimal stomatal conductance and optimal leaf-level $CO_2$ assimilation rate are found as (solid line in Fig. 2A),

$$g_{opt} = \frac{w_0(x_0 - x_T)}{vaDLt_d} \text{ (same as Eq. (11) in the main text)}, \tag{26}$$

$$A_{L,opt} = c_a \left( \frac{1}{k} + \frac{vaDLt_d}{w_0(x_0 - x_T)} \right)^{-1}. \tag{27}$$

Using the optimal stomatal conductance in Eq. (11), the soil water balance of Eq. **Error! Reference source not found.** can be solved to obtain the time trajectory of soil moisture during the dry-down (solid line in Fig. 2B),

$$x = x_0 - vag_{opt}DLt = x_0 - (x_0 - x_T)\frac{t}{t_d}, \tag{28}$$

where, on the right-hand side, it is clear that the optimal solution leads to a linear decrease in soil moisture from the initial

soil moisture $x_0$ to the final value $x_T$. When limited soil moisture constrains water flows, optimal stomatal conductance deviates from the time-invariant value of Eq. (11), leading to a nonlinear decrease in $x$ during a dry period, as explained in OPT3.

**B2. Derivation of OPT3: dynamic feedback optimization with transpiration rate limited by soil moisture**

The decrease in transpiration during drying is often included in soil-plant-atmosphere models through a piecewise linear

function, representing water stress-induced reductions in $E$ (Federer, 1979; Sloan et al., 2021). These observations motivate the inclusion of a further constraint in the optimization relative to OPT1 and OPT2, in the form of a soil moisture-limited transpiration rate under dry conditions that effectively constrains the allowable range of stomatal conductance (Manzoni et al., 2013),

$$E_w = \frac{w_0 \kappa}{v} x. \tag{29}$$

Here, the subscript 'w' refers to water-limited conditions, $v$ adjusts the units so that $E_w$ has the same units as $E$ (i.e., mol $H_2O$

(m$^2$ ground)$^{-1}$ s$^{-1}$), and $\kappa$ is a coefficient with units of d$^{-1}$ that captures the effect of limited rate of water supply from the bulk soil to the roots. For simplicity, $\kappa$ can be approximated as the saturated hydraulic conductivity (m d$^{-1}$) divided by the soil water storage capacity $w_0$ (m). This approximation implies that $E_w$ scales linearly with soil moisture, thus neglecting the nonlinear effect of soil moisture on hydraulic conductivity under unsaturated conditions (Mualem, 1986). Therefore, we expect slower reductions in transpiration as soil dries compared to using a nonlinear relation between $E_w$ and $x$.

Since $E = E_L L = agDL$ (Eq. (1) and (4)) and the water flux through the soil-plant-atmosphere continuum is conserved at the daily (or longer) time scale, we can equate water supply from the soil ($E_w$) and demand by the canopy ($E$), and obtain $E_w = agDL$, where $g$ is different from the optimal value due to the limited water supply from the soil. Solving for $g$ yields the stomatal conductance under water limited conditions (dashed line at low $x$ in Fig. 2c),

$$g_w = \frac{w_0 \kappa}{vaDL} x \text{ (same as Eq. (12) in the main text).} \tag{30}$$

This value of stomatal conductance represents a so-called 'boundary' for the optimization problem. Because the transpiration
rate is a linear function of soil moisture (Eq. (29)), the time trajectory of $x$ in water-limited conditions is found by solving Eq. **Error! Reference source not found.** as (dashed line at $t > t^*$ in Fig. 2b),

$$x_w(t) = x^* e^{-\kappa(t-t^*)}, \tag{31}$$

where $t$ is measured since the beginning of the dry period, and $x^*$ and $t^*$ are respectively the soil moisture and the time at the transition between well-watered and water-limited regimes (open circles in Fig. 2). The stomatal conductance at the transition point is also found by substituting $x = x^*$ in Eq. (12).

Next, we can obtain $x^*$, $t^*$, and $\lambda^*$. Three equations are set up to match the optimal solution under well-watered conditions and the water-limited solution in dry conditions: i) a continuity condition for stomatal conductance; ii) a continuity condition for soil moisture; and iii) a constraint on the amount of soil water left at the end of the dry-period (set at $x_T$ as in OPT2):

i) $\quad g_{opt}(t^*) = g_{opt}^* = k\left(\sqrt{\frac{c_a}{a\lambda^* D}} - 1\right) = \frac{w_0 \kappa}{vaDL} x^*,$ $\tag{32}$

ii) $\quad x(t^*) = x^* = x_0 - \frac{vaDL}{w_0} g_{opt}^* t^*,$ $\tag{33}$

iii) $\quad x_w(t_d) = x_T = x^* e^{-\kappa(t_d - t^*)}.$ $\tag{34}$

The system of Eq. (32)-(34) can be solved to obtain the unknowns $x^*$, $t^*$, and $\lambda^*$ (and thus also $g_{opt}$ for the initial phase at $t < t^*$). To this aim, Eq. (32) and (33) are solved as a function of $t^*$,

$$x^* = \frac{x_0}{1 + \kappa t^*}, \tag{35}$$

$$g_{opt}^* = \frac{x_0 w_0 \kappa}{vaDL(1 + \kappa t^*)} \text{ (same as Eq. (13) in the main text),} \tag{36}$$

whereas the remaining condition in Eq. (34) can be solved numerically for $t^*$ for a given $x^*$ (open circles in Fig. 2). Because optimal $g$ is time-invariant for $t < t^*$, we can also conclude that $g_{opt} = g_{opt}^*$ for any $t$ before the breakpoint $t^*$. This solution of the optimization problem based on the continuity equations at the boundary between well-watered and water limited regimes leads to the same result obtained by adding a Lagrange multiplier within the Hamiltonian to account for the constraint of Eq. (12) (Manzoni et al., 2013).

To summarize the solution of the OPT3 model (dashed lines in Fig. 2), optimal stomatal conductance is initially constant and equal to $g_{opt}^*$ (Eq. (13)), until soil moisture becomes limiting at $x^*$. At this point, stomatal conductance is constrained by water supply from the soil and is given by $g_w$ (Eq. (12)). The more limiting the water supply, the longer the time under water limitation and the higher $g_{opt}^*$ in the initial phase of the dry-down to ensure that all the soil water is used. After calculating stomatal conductance, transpiration and net $CO_2$ assimilation rates are obtained using Eq. (1) and (3) as before.


**Appendix C: Covariation of rooting depth and leaf area index**

In this appendix, we explore the consequences of coordination between rooting depth ($Z_r$), which affects the soil water storage capacity ($w_0$), and leaf area index ($L$) on gas exchange predicted by the stomatal optimization model OPT2. We start by showing theoretical and empirical evidence for relations between $Z_r$ and $L$, and then demonstrate analytically their consequences on optimal stomatal conductance, and thus on net assimilation and transpiration rates.

Aboveground biomass (including leaves) and $Z_r$ co-vary during plant growth, as deeper roots are necessary to acquire soil resources and to stabilize the plant as it grows. To account for this coordinated allocation above- and below-ground, a scaling relation controlled by the exponent $\beta$ can be postulated,

$$Z_r \sim L^\beta. \tag{37}$$

Allometric theory predicts that plant leaf area scales as plant height to power three, and that root extent (lateral and vertical) scales linearly with height (Kempes et al., 2011). It follows that $Z_r$ of an individual plant should scale as leaf area to 1/3, or— for a given plant density—$Z_r$ at the plant population level should scale with $L$ with $\beta = 1/3$. Data from herbaceous vegetation suggests $\beta = 0.40$—significantly higher than 1/3, though numerically close (Fig. C1a). This data was obtained from plants growing over a few months only and without physical limits to root extension. Therefore, this scaling relation can be regarded as an extreme case of coordination between rooting depth and leaf area. However, a shallow bedrock, hard pans, groundwater, or permafrost set physical limits to the vertical extent of roots, suggesting that in adult trees with constrained root extent, $\beta = 0$. Indeed, trends in $Z_r$ with leaf area as tree size (and thus age) increases are not as well defined as for herbaceous vegetation growing in unconstrained soil (Pirtel et al., 2021), and the scaling exponent approaches zero (Fig. C1b). It should be noted that the number of data points for trees is limited, leading to high uncertainty in $\beta$, because most studies on root-leaf coordination compare species rather than following changes in rooting depth and leaf area as trees age. Between these two-end member cases—coordinated rooting depth and leaf area with $\beta \approx 0.4$ *vs.* fixed, physically constrained rooting depth), we expect a range of plausible relations between leaf area index and rooting depth.

Equation (11) shows that the optimal stomatal conductance scales as the ratio of $w_0$ over $L$, where $w_0$ is the product of $Z_r$, soil porosity, and difference in saturation between field capacity and wilting point. Therefore, accounting for the possible coordination of $w_0$ and $L$ via Eq. (37), the leaf area effect on stomatal conductance becomes,

$$g_{opt} \sim \frac{w_0}{L} \sim \frac{Z_r}{L} \sim L^{\beta-1}. \tag{38}$$

This equation indicates that optimal stomatal conductance is inversely related to $L$ (and thus atmospheric $CO_2$ concentration) as long as $\beta < 1$, which is likely the case based on the results shown in Fig. C1. When $\beta = 0$ (i.e., $Z_r$ independent of $L$), the analytical solution used in the main text is recovered. When $\beta$ increases, the effect of higher $L$ on stomatal conductance decreases, which in turn alters the predicted optimal stomatal conductance-atmospheric $CO_2$ concentration relations, as illustrated in Fig. C2. Increasing values of $\beta$ reduce the LAI-mediated negative effect of elevated $CO_2$ on optimal stomatal

conductance and leaf transpiration (Fig. R2A), creates a positive $CO_2$ effect on canopy transpiration (which is insensitive to $CO_2$ concentration when $\beta = 0$) (Fig. R2B), and enhances the positive $CO_2$ effect on both leaf and canopy net assimilation (Fig. R2C-D). In contrast, the positive $CO_2$ effect on water use efficiency is reduced when $\beta > 0$. However, for reasonable values of $\beta$ between 0 and 0.4, the effects on the $CO_2$ responses are minor (green shaded area in Fig. C2), and only for unrealistically high $\beta$ values (e.g., $\beta = 1$, dotted curves in Fig. C2) the response of stomatal conductance becomes flat and
that of canopy transpiration becomes large and positive.

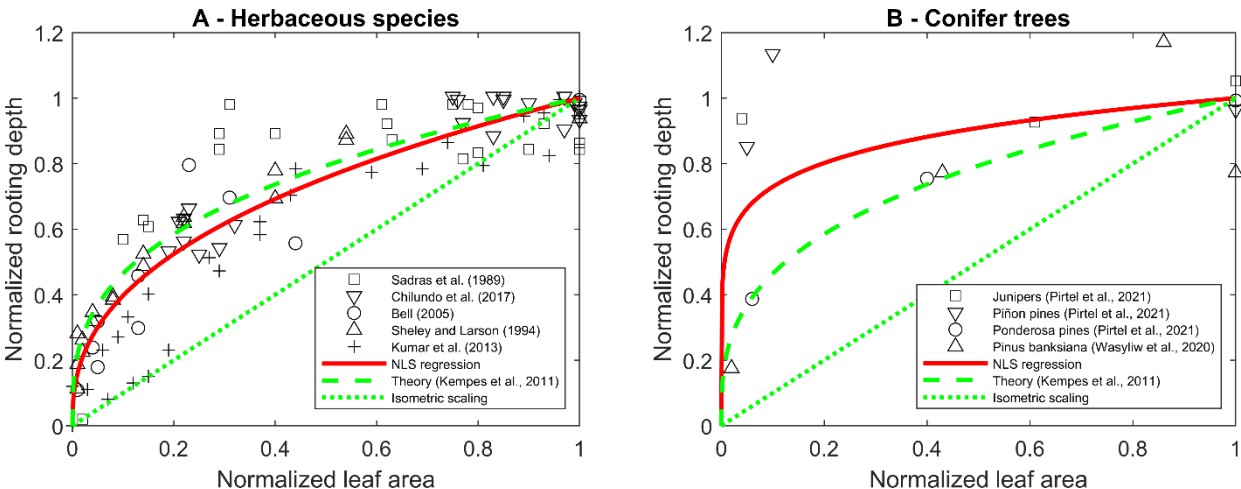

**Fig. C1. Maximum rooting depth as a function of leaf area during plant growth, as measured: A) across herbaceous**
**wild and cultivated species, and B) in four groups of conifer tree species. Both root depth and leaf area are normalized by the maximum values for each species to allow a visual comparison (data from Sadras et al., 1989; Chilundo et al., 2017; Bell, 2005; Sheley and Larson, 1994; Pirtel et al., 2021; Kumar Rohitashw et al., 2013; Wasyliw and Karst, 2020). The red curves are allometric scaling relations obtained through nonlinear least square (NLS) fitting of the data: normalized root depth~normalized leaf area$^\beta$ (A: $\beta$ =0.40 (confidence interval: 0.37-0.44), $R^2$=0.87;**
**B: $\beta$ =0.14 (confidence interval: 0.03-0.25), $R^2$=0.31). The dashed green curves are theoretical scaling relations with $\beta$ =1/3 (Kempes et al., 2011). The dotted green curves represent the unrealistic case of isometric scaling ($\beta$ =1), shown only for reference.**

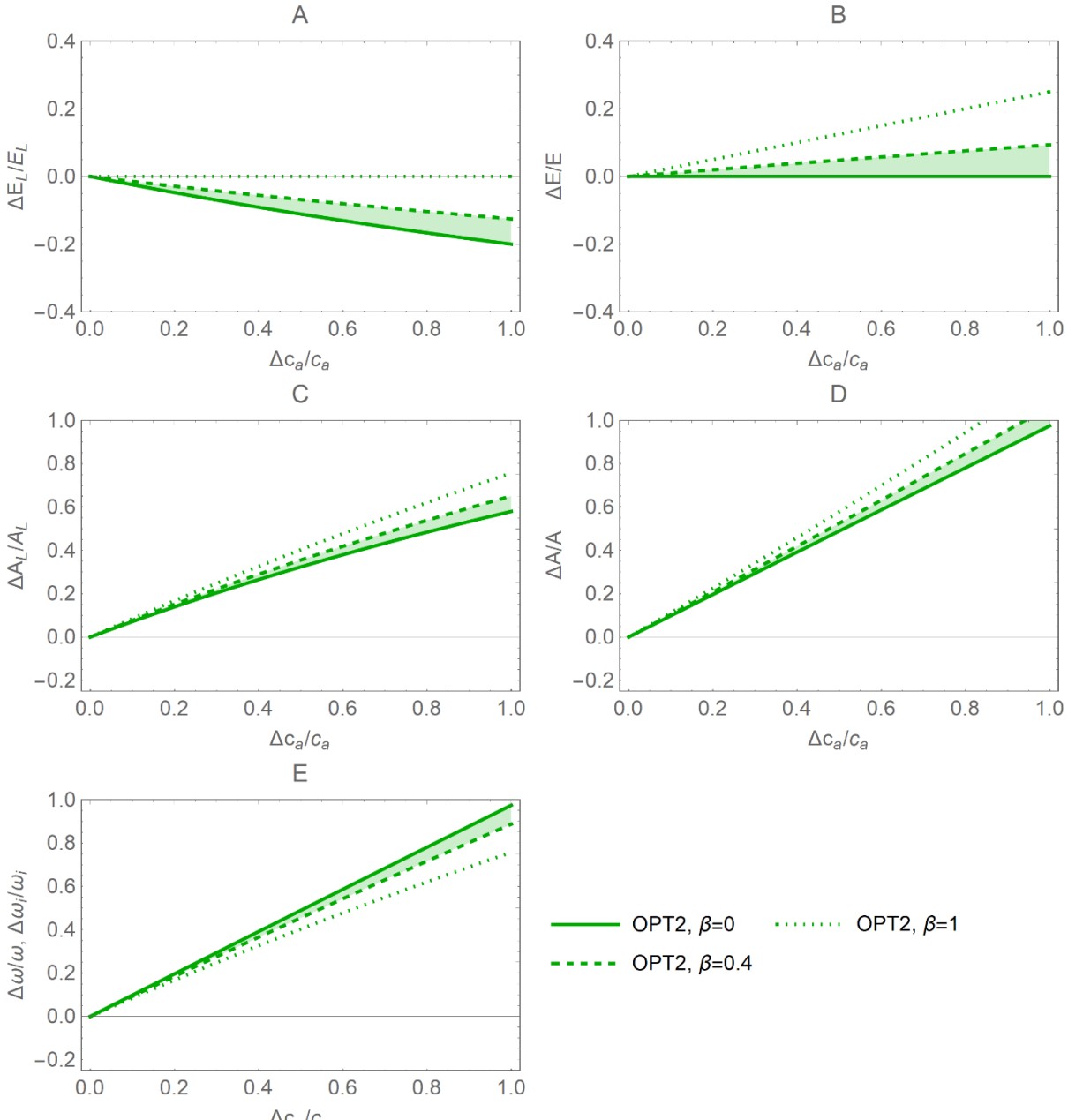

**Fig. C2. Relative changes in leaf-level (A, C) and canopy-level (B, D) gas exchange rates as a function of relative change in atmospheric CO₂ concentration $c_a$, as predicted by the optimal stomatal control model OPT2 for different values of the root depth-leaf area scaling exponent ($\beta$): A) leaf-level transpiration rate ($E_L$), B) canopy-level transpiration rate ($E$), C) leaf-level assimilation rate ($A_L$), D) canopy-level assimilation rate ($A$), and E) water use efficiency ($\omega$). The solid lines correspond to the limiting case of fixed rooting depth ($\beta = 0$); the dashed lines correspond to the empirically-derived scaling exponent $\beta = 0.4$; the dotted lines represent the unrealistic case of isometric scaling ($\beta = 1$). The shaded area between the solid and dashed curves indicates the range of feasible outcomes. Vapor pressure deficit and dry period length are fixed (Table 2); $\alpha = 0.5$.**

## Data availability

Data shown in Fig. 3 are reported in the Supplementary Information.

## Author contributions

SM, GGK, and GV designed the study, with feedback from all co-authors. SM developed the model, produced the results and drafted the manuscript. All co-authors commented on the draft and contributed to the manuscript.

## Competing interests

The authors declare that they have no conflict of interest.

## Special issue statement

This manuscript is submitted to the special issue entitled "Global change effects on terrestrial biogeochemistry at the plant–soil interface".

## Acknowledgements

We thank Stanislaus J. Schymanski for his in-depth comments on an earlier version of the manuscript, and Benjamin Stocker and an anonymous reviewer for insightful comments during the discussion phase. The Discussion also benefitted from discussions with Yair Mau and Yuval Bayer. This project has received funding from the European Research Council (ERC) under the European Union's Horizon 2020 research and innovation programme (grant agreement No 101001608). GV thanks the Swedish Research Council for Sustainable Development FORMAS and the European Commission, grants no. 2018-01820 and 2018-02787. The second grant is in the frame of the collaborative international consortium iAqueduct financed under the 2018 Joint call of the WaterWorks 2017 ERA-NET Cofund; this ERA-NET is an integral part of the activities developed by the Water JPI. XF was supported by National Science Foundation CAREER award DEB-2045610.

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
