# Peer review of "Consistent responses of vegetation gas exchange to elevated atmospheric CO2 emerge from heuristic and optimization models"

_Biogeosciences, 2022_

## Author Comment (AC1)

**Response to Reviewer #1**

We thank Referee #1 for his/her comments, which are addressed as explained below. The Referee's text is reported in *Italic* and our responses in roman.

*This study is a theoretical analysis of the effects of elevated CO2 on carbon and water fluxes at the leaf and canopy level. The authors use four different models describing carbon water relations, one heuristic (PETA) and three optimality based, and run these under different CO2, soil moisture and VPD conditions. Results show that all models predict relatively constant canopy scale transpiration and implicitly an increase in water use efficiency. The paper is well written and the models clearly explained. The question of how plants will respond to future increased atmospheric CO2 and dryer conditions remains one of the critical questions in plant ecophysiology, so in principle this study is highly timely and relevant. However, in practice, it is in my opinion a purely theoretical exercise that largely uses pre-existing models and knowledge and has some pretty big assumptions.*

We thank the reviewer for the supportive comment on presentation and relevance. We agree that this is a theoretical study that leverages existing models, but it is novel in i) comparing PETA and stomatal optimization approaches, ii) exploring the consequences of changes in leaf area index on gas exchange in the optimization models (an often neglected aspect), and iii) analytically demonstrating that previous stomatal optimization models—once the marginal use efficiency is calculated as a part of the optimization problem—are equivalent and can be reduced to a simple mass balance constraint (stomata adjust to use available water). These are novel methodological aspects, but there is a further, more conceptual novelty angle: iv) the contributions of trends in $CO_2$, VPD, and dry period durations on gas exchange and water use efficiency (WUE) are disentangled showing when and how much the positive $CO_2$ effects on net assimilation and WUE are diminished or enhanced by altered VPD and dryness.

Compared to previous work on this topic, our contribution has the advantage of analytical transparency. We agree that the selected models rest on several assumptions (most are common in other modelling studies as well), but we strived to list the assumptions and discussed limitations in result interpretation stemming from these assumptions. In this way, we hope to have achieved a balance between mathematical clarity (which requires more assumptions than in more complex but less transparent models) and realism. Some assumptions have indeed a strong effect on the results, but the analytical framework of both PETA and stomatal optimization models allows us to test scenarios where these assumptions are partly relaxed, as explained in response to the next comment.

*I find that the assumption that most affects the results is that related to the increase in leaf area with increased CO2. The logic behind this assumption is that as photosynthesis increases, there is more carbon available for growth and therefore the leaf area increases in turn. However, from observations and elevated CO2 experiments we know that often there is a shift in*

35 *allocation and while overall growth goes up, leaf area might not, as resources get allocated belowground to deal with other resource limitations. This has implications not only for the ΔL quantity in the model, but also for soil moisture limitations, in particular in the OPT3 formulation. The authors themselves acknowledge this limitation in the discussion and state that optimizing both carbon-water relations and allocation would lead to a too complex model. But surely there are some intermediary options between no belowground allocation and fully optimal allocation. For example, the parameter w0 could*

40 *be varied as it is a function of rooting depth and it could be a trade off between rooting depth and ΔL. Although I am sure the authors know their models better than I do and come up with the best way of doing this.*

The reviewer points to the important issue of resource allocation aboveground (leaf area changes) *vs.* belowground (changes in root density or depth distribution). If rooting density or depth increases under elevated atmospheric $CO_2$ concentration, it

45 is possible that the soil water storage capacity will also increase, providing enough water for a canopy with higher leaf area index. We thank the reviewer for also hinting at a way to assess trade-offs between leaf and root allocation. Indeed, the root zone water storage capacity ($w_0$) depends on rooting depth, as $w_0$ is the product of soil porosity and rooting depth (Table 2 in the manuscript). Letting this parameter vary with changing leaf area index could then account for different degrees of allocation belowground as atmospheric $CO_2$ increases.

50 To proceed in this direction, we need first to prescribe end-member variations in $w_0$. In one extreme scenario, allocation to the roots does not change with elevated $CO_2$ and the associated change in leaf area, because $w_0$ is fixed. Even with higher allocation belowground, if access to water does not change due, e.g., to the presence of a layer limiting root penetration, $w_0$ would also be fixed. In these cases, the results in the submitted manuscript are valid. Empirical data on leaf area-rooting depth coordination are not extensive, as most datasets focus on biomass allocation (e.g., Niklas and Enquist, 2002).

55 Nevertheless, there is evidence that in mature trees within a given plant functional group, higher leaf area does not necessarily translate into deeper roots despite proportional increases of root and leaf biomass (Smith-Martin et al., 2020). This evidence suggests that increasing plant size such as under elevated $CO_2$ could alter both above and belowground biomass (which is expected) but higher leaf area does not imply deeper roots. Elevated $CO_2$ shifts the fine root distribution to deeper layers (Iversen, 2010), but it is not clear if this shift results also in higher water access.

60 In a second scenario, $w_0$ could be coordinated with leaf area index $L$ according to a power law function,

$$w_0 \sim L^\beta, \qquad\qquad\qquad\qquad (1)$$

where the exponent $\beta$ is larger than zero (if $\beta = 0$, Eq. 1 recovers the first scenario) and at most equal to one (isometric scaling). Data in Figure R1 shows that herbaceous annual vegetation exhibits an exponent $\beta < 1$, when considering different time points during plant growth. This suggests that deepening of the root system occurs before substantially increasing leaf

area, but this result is limited to annual species growing over a few months. Any extrapolation to perennial species and trees can be problematic. Nevertheless, the scaling in Figure 3 can be used to define an upper bound for $\beta \approx 0.35$. Most likely, any type of vegetation would exhibit $\beta$ values between these two extreme scenarios, i.e., $0 \leq \beta < 0.35$.

Having established end-member scenarios for rooting depth-leaf area coordination, we can now assess the consequences on predicted gas exchange. Note that in the PETA model there are no parameters related to roots, so the following results are limited to the stomatal optimization model. The combined effect of $w_0$ and $L$ on gas exchange emerges from the relation of optimal stomatal conductance and $w_0$ and $L$ (Eq. (32) in the submitted manuscript; this equation is valid for both OPT2 and OPT3),

$$\bar{g} = \frac{w_0(x_0 - x_T)}{v a D L T_d}. \tag{2}$$

From this equation, it is clear that all else being equal, stomatal conductance is expected to scale as the ratio of $w_0$ over $L$. Accounting for the possible coordination of $w_0$ and $L$ via Eq. (1), the leaf area effect on stomatal conductance becomes,

$$\bar{g} \sim L^{\beta - 1}. \tag{3}$$

This remains an inverse relation qualitatively similar to the one assumed in the submitted manuscript (where $\beta = 0$), as long as $\beta < 1$, which—as discussed above—is likely the case. However, when $\beta$ increases, the effect of higher $L$ on stomatal conductance decreases, which in turn alters the predicted optimal stomatal conductance-atmospheric $CO_2$ concentration relations, as illustrated in Fig. R2. Increasing values of $\beta$ reduce the LAI-mediated negative effect of elevated $CO_2$ on stomatal conductance and leaf transpiration (Fig. R2a), creates a positive $CO_2$ effect on canopy transpiration (which is insensitive to $CO_2$ concentration when $\beta = 0$) (Fig. R2b), and enhances the positive $CO_2$ effect on both leaf and canopy net assimilation (Fig. R2c-d). In contrast, the positive $CO_2$ effect on water use efficiency is reduced when $\beta > 0$.

If offered the opportunity to revise the manuscript, we will introduce the data analysis in Fig. R1 (including also additional datasets we might find) and include Fig. R2 either in the Supplementary Information or in the main text depending on comments from the editors and other reviewers. This analysis will strengthen the discussion points in section 4.4, which will become less speculative and more supported by a dedicated analysis.

[Figure]

**Fig. R1. Maximum rooting depth as a function of leaf area during plant growth, as measured across herbaceous wild and cultivated species. Both root depth and leaf area are normalized by the maximum values for each species to allow a visual comparison (data from Sadras et al., 1989; Chilundo et al., 2017; Bell, 2005; Sheley and Larson, 1994). The red curve is the allometric scaling relation obtained through least square fitting of the data: normalized root depth~(normalized leaf area)$^{0.34}$ ($R^2$=0.89).**

90

[Figure]

**Fig. R2. Relative changes in leaf-level (A, C) and canopy-level (B, D) gas exchange rates as a function of relative change in atmospheric CO₂ concentration $c_a$, as predicted by the optimal stomatal control model OPT2 for different values of the root depth-leaf area scaling exponent ($\beta$): A) leaf-level transpiration rate ($E_L$), B) canopy-level transpiration rate ($E$), C) leaf-level assimilation rate ($A_L$), D) canopy-level assimilation rate ($A$), and E) water use efficiency ($\omega$). The solid lines correspond to the limiting case of fixed rooting depth ($\beta = 0$); the dashed lines correspond to the case shown in Fig. R1; the dotted lines represent the extreme (and unrealistic) case of proportional changes in leaf area and rooting depth ($\beta = 1$). Vapor pressure deficit and dry period length are fixed (Table 2); resource availability $\alpha = 0.5$.**

*Also related to the biomass growth and leaf area question, I find the α parameter somewhat confusing. It is meant to represent resource availability. What this appears to mean in the model, if I understand correctly, is that for a low resource availability (α close to zero), there is a strong increase in leaf area. This is because at high resource availability the canopy would be already almost closed and there is no room to grow while conversely at a low availability the canopy is open and the plants can grow more.*

The reviewer's interpretation is correct. In the publication defining the resource availability index $\alpha$ (Donohue et al., 2017), it is related to leaf area index—if $L$ is large, α is low, indicating that resources are not sufficient to ensure a full canopy. This definition based on leaf area, however, neglects the possibility that maximum leaf area (suggestive of high resource availability) can change depending on location. For example, high-radiation environments can sustain a larger leaf area compared to low-radiation environments. To avoid a this assumed correlation of $L$ and $\alpha$, we thus decided to let these quantities vary independently. It should be noted that except for the initial analysis in Fig. 5, all our results are obtained for fixed $\alpha=0.5$, indicating an intermediate canopy state that permits some degree of leaf area adjustment.

*But if the resource in question is for example nutrients, at low availability we would in reality see no growth response whatsoever, as we do at, for example, the EucFACE experiment. This would make the ΔL dependency on α more like a bell shape. I think this part of the model formulation needs some further explanation and discussion.*

EucFACE has been characterized as a site of intermediate resource availability, with $\alpha=0.57$ (see Table 2 in Donohue et al., 2017). This low value of $\alpha$ is due to the relatively low leaf area index at EucFACE, but the reviewer is correct in saying that low $\alpha$ does not imply high growth potential—in fact, quite the opposite if soil nutrients are limiting. We see the point and share the concern, though for the purpose of our study, we set $\alpha$ to a fixed value in order to study the effect of atmospheric $CO_2$ concentration, VPD and dry period duration on gas exchange. Therefore, a different relation between leaf area index change and $\alpha$ would not alter our results, but it would slightly alter their interpretation—in the original submission, we discussed sensitivities to elevated $CO_2$ in terms of resource availability, while we should instead focus on conditions that allow plants to adjust their leaf area or not. This change in terminology is feasible, and we propose to re-interpret $\alpha$ by focusing on forest age and in general canopy status with respect to a locally-defined maximum leaf area index. These aspects are already mentioned in the current manuscript, but we can explain them further in the Theory section, as well as in the Discussion.

**References**

Bell, L.: Relative growth rate, resource allocation and root morphology in the perennial legumes, Medicago sativa, Dorycnium rectum and D-hirsutum grown under controlled conditions, PLANT SOIL, 270, 199–211, https://doi.org/10.1007/s11104-004-1495-6, 2005.

Chilundo, M., Joel, A., Wesstrom, I., Brito, R., and Messing, I.: Response of maize root growth to irrigation and nitrogen management strategies in semi-arid loamy sandy soil, FIELD CROPS Res., 200, 143–162, https://doi.org/10.1016/j.fcr.2016.10.005, 2017.

Donohue, R. J., Roderick, M. L., McVicar, T. R., and Yang, Y. T.: A simple hypothesis of how leaf and canopy-level transpiration and assimilation respond to elevated $CO_2$ reveals distinct response patterns between disturbed and undisturbed vegetation, J. Geophys. Res.-Biogeosciences, 122, 168–184, https://doi.org/10.1002/2016jg003505, 2017.

Iversen, C.: Digging deeper: fine-root responses to rising atmospheric $CO_2$ concentration in forested ecosystems, NEW Phytol., 186, 346–357, https://doi.org/10.1111/j.1469-8137.2009.03122.x, 2010.

Niklas, K. J. and Enquist, B. J.: On the vegetative biomass partitioning of seed plant leaves, stems, and roots, Am. Nat., 159, 482–497, 2002.

Sadras, V., Hall, A., Trapani, N., and Vilella, F.: Dynamics of rooting and root-length - leaf-area relationships as affected by plant-population in sunflower crops, Field Crops Res., 22, 45–57, https://doi.org/10.1016/0378-4290(89)90088-9, 1989.

Sheley, R. and Larson, L.: Comparative growth and interference between cheatgrass and yellow starthistle seedlings, J. RANGE Manag., 47, 470–474, https://doi.org/10.2307/4002999, 1994.

Smith-Martin, C., Xu, X., Medvigy, D., Schnitzer, S., and Powers, J.: Allometric scaling laws linking biomass and rooting depth vary across ontogeny and functional groups in tropical dry forest lianas and trees, NEW Phytol., 226, 714–726, https://doi.org/10.1111/nph.16275, 2020.

---

## Author Comment (AC2)

**Response to Reviewer #2**

We thank Prof. Stocker for his comments, which are addressed as explained below. The Referee's text is reported in *Italic* and our responses in roman.

*This paper contrasts predictions of contrasting models of ecosystem transpiration and assimilation during rain-free periods ("dry-down events") under different levels of elevated atmospheric CO2 (eCO2), modulated also by simultaneous changes in temperature and relative humidity of the ambient air. The models are formulated in a simplified form to find analytical solutions, while maintaining some essential feedbacks and resolving principles that are funded on observations and*

*understanding of ecosystem mechanisms. This is a different approach compared to the approach taken in global vegetation models and the land components of Earth System Models. The advantage of reduced-form approaches as presented here is that emerging ecosystem behaviour can be linked directly to assumptions and mathematical properties of the model, and thus offer insights into "what matters most".*

Indeed, this is our main goal—capturing with closed form solutions the essential relations between climatic and soil conditions, and gas exchange rates.

*In my understanding, the most important result of the present study is that, irrespective of the chosen model, eCO2-driven acclimation of leaf-level physiology and ecosystem-level structure (morphology) interact in such a way that a reduction of*

*leaf-level transpiration is compensated by increased ecosystem foliage area (L) to fully exploit a constrained (limiting) resource - here water (Abstract l. 29 and Conclusions l., 736). This is a useful insight for understanding observed vegetation greening and streamflow trends in water-limited regions of the Earth and it is interesting that this is a consistent prediction that appears not to be subject to model formulation. I think the paper by Manzoni et al. could be a useful contribution to the literature if the authors manage to convincingly address a few major issues that I would like to raise in the following. I*

*should disclaim that I did not verify the (extensive) algebra presented in the manuscript. Please excuse me for the limited time I can invest in this review.*

We thank Prof. Stocker for his support and constructive criticism, which is addressed below. Our conclusions support the hypothesis that plants allocate resources to exploit limiting water. Another important—albeit methodological rather than conceptual—result is that a full solution (i.e., including solving for the marginal water use efficiency) of the stomatal optimization is required to recover the observed effect of elevated atmospheric $CO_2$ concentration on gas exchange. This finding resolves a long-standing controversy on the use and implementation of stomatal optimization models (as also noted by the reviewer in point (2) below).

*MAJOR*

*(1) While the interaction of physiological and structural adjustments are presented as a key conclusion, it is unclear to what extent this is a model prediction or an assumption. Apparently, changes in L are prescribed to increase under eCO2. This seems to imply that this interaction is not funded on first principles but is a direct reflection of an assumption that is built*
*into all models (although admittedly an assumption with demonstrated empirical support, Fig. 3). Similarly, the conclusion that vegetation physiology and structure adjusts in such a way as to fully exploit the limiting resource, seems to be an assumption, rather than a prediction. The Cowan & Farquhar (1977) approach to predicting physiological responses subject to A-λE maximisation starts with the presumption of a constrained amount of plant-accessible water and therefore has to predict that this amount of water be transpired over the course of a dry-down ("imposing the condition that all*
*available water is used by the end of the dry period", l. 670).*

Regarding leaf area changes, they are assumed to be a function of atmospheric $CO_2$ concentration for all the models. In this sense, the models do not predict changes in leaf area, but changes in gas exchange after accounting for prescribed variations in leaf area. The rationale for this choice is that leaf area varies over time scales longer than those typical of stomatal
responses to environmental fluctuations (including soil moisture), so that we can 'separate the temporal scales' of the two problems—stomatal regulation (a variable to be solved for) and leaf area adjustment (assumed). It is also correct that we assumed that plants consume soil available water during a 'typical' dry period. (We comment below in the response to another comment about plants 'knowing' how long a dry period is.) This assumption sets only a boundary for the optimization problem, but does not offer clues on the trajectory of stomatal conductance during the dry period—that is the
actual solution of the optimization problem. As it turns out, stomatal closure during a dry period occurs only when transpiration becomes water supply limited, otherwise a time-invariant soil moisture throughout the dry period represent the optimal trajectory. This is also the only trajectory that maximizes the cumulative net assimilation over that period. These trajectories (time-invariant or decreasing stomatal conductance) are not assumed, but are the result of the optimization problem solution.

*Similarly for the PETA model (as I understand Donohue et al., 2013 - please clarify if not the same principles are applied in Donohue et al., 2017): Foliage area changes are predicted subject to "known" changes in leaf-level water use efficiency and a constrained amount of available water (corresponding to precipitation). In other words, all available water is assumed to be consumed. (But then, I must be misunderstanding something here, given that L changes are prescribed also in*
*the PETA model here.) A clarification of these points would help to better understand the relevance of assumptions and the*

*purely predictive ability of the model (and therefore what can defensibly be presented as a conclusion from the present study).*

Also according to our interpretation, in the PETA model leaf area changes depend on how leaf-level water use efficiency varies with atmospheric $CO_2$ concentration and vapor pressure deficit (VPD). However, there are some differences between the approaches by Donohue et al. (2013) and Donohue et al. (2017). In the earlier paper, long-term canopy transpiration was assumed to be constrained by a fixed amount of precipitation, which implies $dE_l/E_l = -dL/L$ (using the notation in Table 1 of the submitted manuscript; $E_L$: leaf-level transpiration rate, $L$: leaf area index). With this constraint, leaf area changes can be directly related to net assimilation, VPD, and atmospheric $CO_2$ concentration changes, which allows partitioning variations in leaf area among the different climatic drivers. In the later paper, the resource availability index $\alpha$ is introduced, and leaf area is expressed as an empirical function of $\alpha$ without a specific constraint on canopy transpiration. In fact, their Figure 1 shows that canopy transpiration varies mildly with $\alpha$ (Donohue et al., 2017). This is the PETA formulation we used; therefore, also in our contribution canopy transpiration is not assumed to be fixed (e.g., see variations in canopy transpiration with increasing atmospheric $CO_2$ in our Figure 5b).

We agree with the reviewer that the differences between model assumptions vs. predictions were not presented clearly and propose to revise the manuscript in the following ways:

- Improved Figure 1, where we can spell out more clearly what are the assumptions, and what are the results of each model (Figure R1).
- Added clarifications in Section 2.2 where describing the PETA model: "Equation (8) also shows that canopy transpiration can vary unless both leaf-level transpiration and leaf area index are constant. This result of the PETA model differs from a key assumption of the stomatal optimization model (Sections 2.3.2 and 2.3.3)." and in Section 2.3.2 where describing the solution of the stomatal optimization problem: "It is important to emphasize that this specific stomatal conductance trajectory is not a result of our assumption that all available water is used—it is rather the solution that best balances the water consumption rate over time to maximize net assimilation."

[Figure]

**Figure R1. Revised conceptual representation of the models used to assess gas exchange responses.**

*(2) Another potentially useful insight of the present study is that realistic stomatal responses to eCO2 can be predicted only*
*with stomatal optimality models with a "dynamic feedback" (OPT2 and OPT3), but not with an "instantaneous optimality"*
*model (OPT1). In the models investigated here, this "instantaneous optimality"-based prediction follows from a constant λ,*
*and given a constrained amount of transpirable water. This result is then used to argue that OPT1 (representing in general*
*an "instantaneous optimality" model) is not realistic. This could be misunderstood as a demonstration of a general*
*uselessness of similar "instantaneous optimality"-based models (e.g., Prentice et al., 2014; Sperry et al., 2016; Wolf et al.,*
*2016). However, not all "instantaneous optimality" models are based on optimising C assimilation for a constrained amount*
*of transpiraition, and stomatal conductance is (in line with observations) predicted by several "instantaneous optimality"*
*models to decline with rising CO2 (Stocker et al., 2020, see their Eq. C1; Joshi et al., 2021 Fig. 2). Hence, the presentation*
*of instantaneous vs. dynamic feedback optimality models runs the danger of creating a straw-man argument. A clarification*

*and intuitive explanation of why stomatal conductance is predicted to increase by OPT1 but not in other "instantaneous*

*optimality"-based models, seems needed.*

We totally agree. There is value in instantaneous optimization approaches (which should perhaps be called 'instantaneous maximization' since no optimal trajectories are determined), though it might be important to remind readers that in general instantaneous optimization does not guarantee that organism fitness is maximized over a specified time frame (dry down as in our case or more in general life span). In other words, using resources now at very high rate might maximize gains today, but result in stress and possibly higher mortality tomorrow. The problem is even more complicated when competition among mutants or species is considered—but even in that case a 'local' maximization does not guarantee an optimal (or evolutionary stable) outcome. Yet, models defining instantaneous gains and costs and maximizing the difference between these two terms can be highly effective and could represent a pragmatic solution for vegetation modelling (though not an

'optimal' one in a mathematical sense).

If encouraged to revise our manuscript, we can provide a more balance comparison with other approaches (including acknowledgement of their value), starting by clearly stating in the Discussion (Section 4.4) that: "For example, the optimization model based on Prentice et al. (2014) correctly predicts the observed short-term decrease in stomatal conductance under elevated atmospheric $CO_2$ (Eq. (C1) in Stocker et al., 2020) without invoking leaf area adjustments."

Moreover, we can be more comprehensive in the Introduction, referring to synthesis papers where optimization models are compared and including a statement to explain that in this manuscript we focus on a specific type of optimization-based models—those formulated as an optimal control problem (see our response to a minor comment below).

Regarding an intuitive explanation of the counterintuitive optimal response of stomatal to elevated CO2 concentration with the OPT1 formulation, we can explain in these terms (Discussion Section 4.4): "These erroneous responses arise because at low $CO_2$ concentration a small increase in stomatal conductance results in large net $CO_2$ assimilation gains compared to the higher water losses, resulting in the counterintuitive opening of stomata as atmospheric $CO_2$ concentration is increased."

*(3) The discussion of the investigated models in the context of the extensive literature on other modelling approaches to simulating physiology in response to soil moisture dry-downs is relatively slim. The single statement referring to such*

*alternatives on l. 680 ("While these approaches are more physiologically accurate and their predictions compare well with observed trends, they do not guarantee that the water use is optimal over the whole optimization period.") does not do it justice in my view. I want to avoid a more fundamental debate over the "constrained water" assumption of Cowan & Farquhar (1977) (How can a plant know in advance how long the current dry-down will last? How can it know how to optimally make use of available water from now until the [future] end of the dry-down? Why wouldn't it be advantageous for*

*a competitor to consume water immediately rather than save it for the future?), but the justification of not discussing alternative modelling approaches by stating that "they do not guarantee that the water use is optimal over the whole optimization period" seems unfair - particularly in view of the argument that I want to avoid getting into ;-).*

We see the point of the reviewer, and as also mentioned above, we recognize that a more balanced discussion is warranted.

This is entirely feasible in the Discussion without altering significantly the structure, mostly expanding Section 4.4.

Regarding the fundamental question of the time scale over which plants can be expected to maximize carbon gains, we can be more open in a revised manuscript. We assumed that this time scale corresponds to an 'average' dry down duration, which can be criticised. Assuming that plants maximize carbon gains instantaneously is equally problematic in our view, because it implies that plants 'know' that saving resources today is not particularly useful (whereas it might be). In a way, with all optimization or maximization approaches, we are trying to reverse-engineering how evolution programmed plants to respond to environmental changes. Plants cannot foresee the future, but it is fair to assume that their responses are tuned to the environmental where they live, which—as a first order approximation—can be represented by the average conditions experienced over several generations. On these grounds, a dry down duration or the expectation that net gain maximization today will not harm the plant tomorrow are equally strong—and equally reasonable—assumptions. Depending on environmental conditions and competition patterns, one or the other might prove more useful for prediction.

*(4) The present manuscript is heavy on algebra. I understand that this is central to the reasoning of the presented analysis, but I recommend that all efforts be made that this manuscript can be read and its reasoning intuitively understood without deciphering the algebra. In general, reasonable efforts should be made to reduce the algebra, possibly relegating parts to*

*the Appendix, while still maintaining the essential descriptions. Sorry that my point here is not more specific, but I recommend that the presentation of the science be presented to appeal to the widest possible audience.*

The methodological section with heavier mathematical derivations is Section 2.3 "Optimal stomatal control models". We can move most of the material from that section to an Appendix, to leave a streamlined text only presenting the results in an intuitive way. Following this approach, we would leave the equations for optimal stomatal conductance in the main text, together with their explanations; the actual mathematical derivations would become Appendix B. Figure 2 would remain in the main text as it illustrates the meaning of the optimal stomatal conductance solution. We would leave instead Sections 2.1 and 2.2 as they are, since they provide basic equations or final results (the latter for the PETA model). With these changes, only five equations would remain in Section 2.3, facilitating the reading of the Methods for a wide audience.

*MINOR*

*The specific scientific question and scope of the manuscript and the model investigation is not immediately clear. The last sentence of the abstract points to the essence being the coordination of physiology and morphology in their response to*

*eCO2. Then, the question is stated more precisely as "... but it is not clear if and under which conditions these two effects balance out." (l. 44). Is this the central question? Does the paper answer this question? If so, could an answer to that*

*question be given more clearly in the Conclusion section? The introduction shifts attention to other questions (l. 65, l. 105-107) and answers to these make up much of the Conclusions section instead. This comes at the cost of a not so well-defined scope overall.*

The ambiguity probably derived from the double scope of this contribution. On the one hand, we highlight some implications of the optimization model that have been previously overlooked (a methodological issue); on the other hand—with both PETA and optimization models—we address the scientific question of how plants respond to elevated atmospheric $CO_2$. Both aspects are relevant. In the submitted abstract we state our central question (formulated as a research gap): "The net effect of elevated $CO_2$ on leaf- and canopy-level gas exchange thus remains unclear." The concluding sentence of the abstract answers this question: "coordination of physiological and morphological characteristics in vegetation to maximize resource use (here water) under altered atmospheric conditions." The same question is brought up at the beginning of the Introduction: "This increase in the canopy-level evaporating surface area could counterbalance the reduction in transpiration caused by stomatal closure at the leaf level, but it is not clear if and under which conditions these two effects balance out." However, we agree with the reviewer that later in the Introductions the questions seem to have changed. We propose to re- phrase them as follows, including a statement of our further methodological aim to compare model predictions and discuss the limitations of the PETA and optimization models:

"1. How do physiological (stomatal conductance) and morphological (leaf area) adjustments combine to determine leaf and canopy gas exchange rates under atmospheric $CO_2$ concentrations?

2. How do physiological and morphological adjustments determine gas exchange responses to combined changes in $CO_2$

concentration and atmospheric or soil drought?

By comparing the predictions of the PETA and optimization models, we provide a theoretical perspective on these questions, while also identifying advantages and limitations in these different modelling approaches."

In the conclusions, we can more clearly answer these questions: "Both models predict low sensitivity of canopy transpiration rates to a changing climate, indicating that morphological adjustments (leaf area increase) compensate physiological adjustments (stomatal closure)." The very last sentence in the Conclusions re-states the same message in slightly different terms: "Overall, these results imply that physiological and morphological traits acclimate to changing environmental conditions in a coordinated manner to ensure that limiting resources such as water are used efficiently." We hope that with these changes, the overall scope is clearer.

*The different model variants could be better linked with specific hypotheses about controls and mechanisms determining stomatal responses to eCO2. Are there specific questions to be answered by comparing predictions from the different models?*

Perhaps the main conceptual difference regarding $CO_2$ responses is between optimization model OPT1 and the other two variants, OPT2 and OPT3. The former should allow capturing short-term responses for fixed $\lambda$ (but it does not, as discussed in the manuscript), whereas the latter two should be able to capture long term responses that include morphological changes (because $\lambda$ varies with atmospheric $CO_2$ via changes in leaf area). The differences between these model variants are presented in Fig. 3 and commented upon in the Discussion. Due to the nature of the optimization approach—in practice a way to avoid a description of detailed physiological processes thanks to an ecological 'goal' function—it is difficult to link these model variants to specific mechanistic hypotheses, so we do not have a clear answer to this comment.

*l. 39 "stimulates plant growth and thus increases leaf area": Is increased leaf area a consequence of stimulated growth?*
Yes, that seems to be the case. Higher net assimilation provides resources for faster growth, and hence also higher leaf area (assuming resources are invested both above and belowground).

*l. 41 "open canopies": A dependence of the eCO2 effect on leaf area subject to initial leaf area (open canopy) is mentioned throughout the manuscript. In view of canopies being open due to water limitation, nutrient limitations, low temperatures, or simply due to young age, is often not specified, but may be relevant for responses and certainly for underlying mechanisms. Could references to "open canopies" be made more specific throughout?*
This is a good point, also raised by Reviewer #2 in relation to the meaning of the resource availability index $\alpha$. Essentially,
in the PETA formulation by Donohue et al. (2017), $\alpha$ represents how much an ecosystem can increase its leaf area index in response to elevated atmospheric $CO_2$. As the reviewer points out, why a canopy is open is important to understand its potential leaf area changes, and to interpret our results (though results are based on a fixed $\alpha$ value, and so are not affected by how we define $\alpha$). To clarify, we propose to interpret $\alpha$ as a measure of forest age ($\alpha$ increasing with age) or canopy status with respect to the maximum leaf area index expected for a site ($\alpha$ decreasing if the canopy is close to its maximum LAI).
Clarifications to this regard can be added in Section 2.2: "This index represents how far vegetation is from the maximum $L$ expected for that location—high $\alpha$ indicates an old stand or in general a stand with $L$ close to the maximum, where additional leaf area increases are not possible (see also Sect. **Error! Reference source not found.**)."

*l. 73-74: "The model is based ..." Add: ... and the assumption that vegetation in water-limited regions makes full use of a*
*constrained flux of water (~precipitation).*
Only the 2013 formulation of the PETA hypothesis relies on this assumption, and we use the 2017 version, so it might be confusing to include this assumption here. Please see also our response to comment (1) above.

*l. 83: "Stomatal optimization" models are referred to as models relying on the "Lagrange multiplier" $\lambda$. This seems to be a*
*too narrow definition of "stomatal optimization". In my understanding, models that predict stomatal responses to changes along the soil-plant-atmosphere continuum may be considered here too (e.g., Sperry et al., 2016; Wolf et al., 2016).*
To address this comment, we can cite other examples, even though they do not perform a formal optimization through time. To avoid a long list of references, while also clarifying our focus, we would add in the Introduction: "Among the numerous optimization-based models available (Mencuccini et al., 2019; Wang et al., 2020, and references therein), we focus here on those formulated as an optimal control problem in which stomatal conductance is solved through time."

*Table 1: Confusing use of 'T' in T_a and T_d, while the two 'T' are different variables with different units. E_SR not explained.*

We can change the symbol used for the dry period duration to $t_d$; we apologize for the confusion on the symbol $E_{SR}$, which appeared by mistake as it was changed (not everywhere apparently) to $E_W$.

*l. 147: Spell out that ci:ca is assumed to remain constant under eCO2.*

This is only an approximation for the denominator of the $A$-$c_i$ curve, not an assumption. In other words, the $A$-$c_i$ curve is linearized, without losing the dependence of $c_i$:$c_a$ on atmospheric $CO_2$ concentration (Eq. (10) in Katul et al., 2010). A clarification can be added by explaining: "As a result, $A_L$ is a linear function of $c_i$, but the slope of the relation decreases with increasing atmospheric $CO_2$ concentration; moreover, this approximation allows retaining variations in $c_i/c_a$ with $c_a$ (Katul et al., 2010)."

*l. 150: "A_L is a linear function of ci but with a declining slope at high CO2 concentration". This seems to be a contradiction in itself. Either it's linear or has a varying slope.*

The slope of the linearized $A$-$c_i$ curve depends on $c_a$, so it can change as atmospheric $CO_2$ increases, but this was not clear in the way we had formulated the sentence. Combining this and the previous comment, we propose to amend this sentence as: "As a result, $A_L$ is a linear function of $c_i$, but the slope of the relation decreases with increasing atmospheric $CO_2$ concentration; moreover, this approximation allows retaining variations in $c_i/c_a$ with $c_a$ (Katul et al., 2010)."

**References**

Donohue, R. J., Roderick, M. L., McVicar, T. R., and Farquhar, G. D.: Impact of CO2 fertilization on maximum foliage cover across the globe's warm, arid environments, Geophys. Res. Lett., 40, 3031–3035, https://doi.org/10.1002/grl.50563, 2013.

Donohue, R. J., Roderick, M. L., McVicar, T. R., and Yang, Y. T.: A simple hypothesis of how leaf and canopy-level transpiration and assimilation respond to elevated CO2 reveals distinct response patterns between disturbed and undisturbed vegetation, J. Geophys. Res.-Biogeosciences, 122, 168–184, https://doi.org/10.1002/2016jg003505, 2017.

Joshi, J., Stocker, B. D., Hofhansl, F., Zhou, S., Dieckmann, U., and Prentice, I. C.: Towards a unified theory of plant photosynthesis and hydraulics, bioRxiv, 2020.12.17.423132, https://doi.org/10.1101/2020.12.17.423132, 2021.

Katul, G., Manzoni, S., Palmroth, S., and Oren, R.: A stomatal optimization theory to describe the effects of atmospheric CO2 on leaf photosynthesis and transpiration, Ann. Bot., 105, 431–442, 2010.

Mencuccini, M., Manzoni, S., and Christoffersen, B.: Modelling water fluxes in plants: from tissues to biosphere, New Phytol., 222, 1207–1222, https://doi.org/10.1111/nph.15681, 2019.

Prentice, I. C., Dong, N., Gleason, S. M., Maire, V., and Wright, I. J.: Balancing the costs of carbon gain and water transport: testing a new theoretical framework for plant functional ecology, Ecol. Lett., 17, 82–91, https://doi.org/10.1111/ele.12211, 2014.

Stocker, B., Wang, H., Smith, N., Harrison, S., Keenan, T., Sandoval, D., Davis, T., and Prentice, I.: P-model v1.0: an optimality -based light use efficiency model for simulating ecosystem gross primary production, Geosci. MODEL Dev., 13, 1545–1581, https://doi.org/10.5194/gmd-13-1545-2020, 2020.

Wang, Y., Sperry, J. S., Anderegg, W. R. L., Venturas, M. D., and Trugman, A. T.: A theoretical and empirical assessment of stomatal optimization modeling, New Phytol., n/a, https://doi.org/10.1111/nph.16572, 2020.

---

## Author Response (AR1)

**Response to the Editor**

We thank Dr. Solly for encouraging us to submit a revised manuscript. We have already detailed how we planned to revise our manuscript in the responses to the reviewers' comments posted in the public discussion forum. In this letter, we only describe how the planned changes have been implemented in the revised manuscript, and leave our arguments in support to those changes in the open discussion to avoid unnecessary repetitions. All comments by the Reviewers are reported in *Italic* and our responses are in roman. Please note that the attached track-change version is not accurate as it had to be prepared *a posteriori* and the 'compare' function in Word marked as modified some text (especially in the Introduction) that in fact had not been altered.

**Response to Reviewer #1**

*This study is a theoretical analysis of the effects of elevated CO2 on carbon and water fluxes at the leaf and canopy level. The authors use four different models describing carbon water relations, one heuristic (PETA) and three optimality based, and run these under different CO2, soil moisture and VPD conditions. Results show that all models predict relatively*

*constant canopy scale transpiration and implicitly an increase in water use efficiency. The paper is well written and the models clearly explained. The question of how plants will respond to future increased atmospheric CO2 and dryer conditions remains one of the critical questions in plant ecophysiology, so in principle this study is highly timely and relevant. However, in practice, it is in my opinion a purely theoretical exercise that largely uses pre-existing models and knowledge and has some pretty big assumptions.*

No change was made in response to this initial supportive comment. We acknowledge that this is a theoretical study whose assumptions are stated as clearly as possible (see additional explanations in the response to the reviewer).

*I find that the assumption that most affects the results is that related to the increase in leaf area with increased CO2. The*

*logic behind this assumption is that as photosynthesis increases, there is more carbon available for growth and therefore the leaf area increases in turn. However, from observations and elevated CO2 experiments we know that often there is a shift in allocation and while overall growth goes up, leaf area might not, as resources get allocated belowground to deal with other resource limitations. This has implications not only for the ΔL quantity in the model, but also for soil moisture limitations, in particular in the OPT3 formulation. The authors themselves acknowledge this limitation in the discussion and state that*

*optimizing both carbon-water relations and allocation would lead to a too complex model. But surely there are some intermediary options between no belowground allocation and fully optimal allocation. For example, the parameter w0 could be varied as it is a function of rooting depth and it could be a trade off between rooting depth and ΔL. Although I am sure the authors know their models better than I do and come up with the best way of doing this.*

We fully agree with the reviewer and introduced a new analysis to address their comment. We collected empirical evidence of the relation between rooting depth and leaf area index and in herbaceous vegetation and trees, focusing on changes in these plant characteristics during growth rather than along spatial gradients. This dataset, together with analytical derivations based on allometric theory, allowed defining a power law relation between rooting depth and leaf area index that generalized our original results to cases in which rooting depth increases with leaf area. This new analysis is included in Appendix C, which also contains a figure illustrating the data and the scaling relations, as well as a variant of main text Fig. 5 where the effect of deeper roots under elevated $CO_2$ concentrations is shown. We conclude from this analysis that for reasonable values of the power law exponent in the rooting depth-leaf area index relation, the effects on the $CO_2$ responses of gas exchange rates are minor and would not alter the main results originally presented. These results are discussed in Section 4.4.

*Also related to the biomass growth and leaf area question, I find the α parameter somewhat confusing. It is meant to represent resource availability. What this appears to mean in the model, if I understand correctly, is that for a low resource availability (α close to zero), there is a strong increase in leaf area. This is because at high resource availability the canopy would be already almost closed and there is no room to grow while conversely at a low availability the canopy is open and the plants can grow more. But if the resource in question is for example nutrients, at low availability we would in reality see*

*no growth response whatsoever, as we do at, for example, the EucFACE experiment. This would make the ΔL dependency on α more like a bell shape. I think this part of the model formulation needs some further explanation and discussion.*

To address the reviewer's concern (shared with Reviewer #2), we re-interpreted α by focusing on forest age and in general canopy status with respect to a locally-defined maximum leaf area index. The updated definition reads "This index represents how far vegetation is from the maximum *L* expected for that location. High α indicates an old stand or in general a stand with *L* close to the maximum, where additional leaf area increases are not possible" (Section 2.2). The most important results of our work (Fig. 6-8) are obtained for a fixed value of α, or in other words for a given canopy status. Therefore, different interpretations on the meaning of α should not alter our conclusions. We do agree with the reviewer that applying the PETA model in a prognostic way across sites would require a consistent estimation approach for α (but that is outside our scope).

**Response to Reviewer #2**

*This paper contrasts predictions of contrasting models of ecosystem transpiration and assimilation during rain-free periods ("dry-down events") under different levels of elevated atmospheric CO2 (eCO2), modulated also by simultaneous changes*

*in temperature and relative humidity of the ambient air. The models are formulated in a simplified form to find analytical solutions, while maintaining some essential feedbacks and resolving principles that are funded on observations and*

*understanding of ecosystem mechanisms. This is a different approach compared to the approach taken in global vegetation models and the land components of Earth System Models. The advantage of reduced-form approaches as presented here is that emerging ecosystem behaviour can be linked directly to assumptions and mathematical properties of the model, and*

*thus offer insights into "what matters most". In my understanding, the most important result of the present study is that, irrespective of the chosen model, eCO2-driven acclimation of leaf-level physiology and ecosystem-level structure (morphology) interact in such a way that a reduction of leaf-level transpiration is compensated by increased ecosystem foliage area (L) to fully exploit a constrained (limiting) resource - here water (Abstract l. 29 and Conclusions l., 736). This is a useful insight for understanding observed vegetation greening and streamflow trends in water-limited regions of the*

*Earth and it is interesting that this is a consistent prediction that appears not to be subject to model formulation. I think the paper by Manzoni et al. could be a useful contribution to the literature if the authors manage to convincingly address a few major issues that I would like to raise in the following. I should disclaim that I did not verify the (extensive) algebra presented in the manuscript. Please excuse me for the limited time I can invest in this review.*

No change was made in response to this initial supportive comment.

*MAJOR*

*(1) While the interaction of physiological and structural adjustments are presented as a key conclusion, it is unclear to what*

*extent this is a model prediction or an assumption. Apparently, changes in L are prescribed to increase under eCO2. This seems to imply that this interaction is not funded on first principles but is a direct reflection of an assumption that is built into all models (although admittedly an assumption with demonstrated empirical support, Fig. 3). Similarly, the conclusion that vegetation physiology and structure adjusts in such a way as to fully exploit the limiting resource, seems to be an assumption, rather than a prediction. The Cowan & Farquhar (1977) approach to predicting physiological responses*

*subject to A-λE maximisation starts with the presumption of a constrained amount of plant-accessible water and therefore has to predict that this amount of water be transpired over the course of a dry-down ("imposing the condition that all available water is used by the end of the dry period", l. 670). Similarly for the PETA model (as I understand Donohue et al., 2013 - please clarify if not the same principles are applied in Donohue et al., 2017): Foliage area changes are predicted subject to "known" changes in leaf-level water use efficiency and a constrained amount of available water (corresponding*

*to precipitation). In other words, all available water is assumed to be consumed. (But then, I must be misunderstanding something here, given that L changes are prescribed also in the PETA model here.) A clarification of these points would help to better understand the relevance of assumptions and the purely predictive ability of the model (and therefore what can defensibly be presented as a conclusion from the present study).*

We agree that the models do not predict changes in leaf area, but changes in gas exchange after accounting for prescribed variations in leaf area. This assumption sets only a boundary for the optimization problem, but does not offer clues on the trajectory of stomatal conductance during the dry period—that is the actual solution of the optimization problem. We used the PETA model formulation from Donohue et al. (2017), in which leaf area is expressed as an empirical function of $\alpha$ without a specific constraint on canopy transpiration (Eq. (7) in the main text); therefore, also in our contribution canopy transpiration is not assumed to be fixed—the finding that transpiration is almost insensitive to elevated atmospheric $CO_2$ concentration is a result of the model. To clarify, we re-phrased the second key assumption stated at the beginning of Section 2: "[plants] have acclimated to the atmospheric conditions by varying their growing season LAI (which is prescribed in both models) and stomatal conductance". This assumption is also re-stated in Sections 2.4: "… the same LAI changes are included in both PETA and optimization models by combining Eq. **Error! Reference source not found.** and **Error!**

**Reference source not found.** to determine $\Delta L/L$" and 2.5: "In both the PETA and optimization models, LAI varies with atmospheric $CO_2$ concentration and VPD in the same manner (Fig. 1)."

We agree with the reviewer that the differences between model assumptions vs. predictions were not presented clearly and revised the manuscript in the following ways:

- In the revised Fig. 1, we presented more clearly the distinction between assumptions and results of each model.
- We clarified in Section 2.2 where describing the PETA model: "Equation (8) also shows that canopy transpiration can vary unless both leaf-level transpiration and leaf area index are constant. This result of the PETA model differs from a key assumption of the stomatal optimization model (Sections 2.3.2 and 2.3.3)." and in Section 2.3.2 where describing the solution of the stomatal optimization problem: "It is important to emphasize that this specific stomatal conductance trajectory is not a result of our assumption that all available water is used. Rather, it is the solution that best balances the water consumption rate over time to maximize net assimilation."

*(2) Another potentially useful insight of the present study is that realistic stomatal responses to eCO2 can be predicted only with stomatal optimality models with a "dynamic feedback" (OPT2 and OPT3), but not with an "instantaneous optimality"*

*model (OPT1). In the models investigated here, this "instantaneous optimality"-based prediction follows from a constant λ, and given a constrained amount of transpirable water. This result is then used to argue that OPT1 (representing in general an "instantaneous optimality" model) is not realistic. This could be misunderstood as a demonstration of a general uselessness of similar "instantaneous optimality"-based models (e.g., Prentice et al., 2014; Sperry et al., 2016; Wolf et al., 2016). However, not all "instantaneous optimality" models are based on optimising C assimilation for a constrained amount*

*of transpiration, and stomatal conductance is (in line with observations) predicted by several "instantaneous optimality" models to decline with rising CO2 (Stocker et al., 2020, see their Eq. C1; Joshi et al., 2021 Fig. 2). Hence, the presentation of instantaneous vs. dynamic feedback optimality models runs the danger of creating a straw-man argument. A clarification*

*and intuitive explanation of why stomatal conductance is predicted to increase by OPT1 but not in other "instantaneous optimality"-based models, seems needed.*

We totally agree, and now explained in the Introduction that we focus on stomatal conductance models based on optimal control; we also provided a more balanced and accurate comparison with other approaches in the Discussion, including acknowledgement of their value:

- Introduction, added citation of Buckley and Schymanski (2014): "This approach is equivalent to performing an
'instantaneous' optimization without considering the soil water dynamics or changes in leaf area that can feedback to leaf-gas exchange, albeit at longer time scales compared to the opening and closure of stomata in response to environmental stimuli (Buckley and Schymanski, 2014)"

- Introduction: "Among the numerous models available (Mencuccini et al., 2019; Wang et al., 2020, and references therein), we focus here on those formulated as an optimal control problem in which stomatal conductance is solved
through time."

- Section 4.4: "Interestingly, also optimizing $c_i/c_a$ to maximize carbon gains minus water transport costs per unit of net $CO_2$ assimilation (Prentice et al., 2014) results in increasing stomatal conductance with $c_a$ at pre-industrial $c_a$ values (Fig. S2 in Joshi et al., 2022)."

- Later in Section 4.4: "Other models based on instantaneous maximization of C gains for given costs offer
alternative frameworks to predict responses to atmospheric $CO_2$ concentrations and other environmental changes (Sperry et al., 2017; Mencuccini et al., 2019; Huang et al., 2018; Bassiouni and Vico, 2021; Prentice et al., 2014; Joshi et al., 2022). For example, the model based on Prentice et al. (2014) correctly predicts the observed short-term decrease in stomatal conductance under elevated (but not pre-industrial) atmospheric $CO_2$ (Eq. (C1) in Stocker et al., 2020) without invoking leaf area adjustments."

Regarding an intuitive explanation of the counterintuitive optimal response of stomatal to elevated $CO_2$ concentration with the OPT1 formulation, we now explained (Discussion Section 4.4): "In the stomatal optimization models, these erroneous responses arise because at low $CO_2$ concentration a small increase in stomatal conductance results in large net $CO_2$ assimilation gains compared to the higher water losses, resulting in the counterintuitive opening of stomata as atmospheric
$CO_2$ concentration is increased."

*(3) The discussion of the investigated models in the context of the extensive literature on other modelling approaches to simulating physiology in response to soil moisture dry-downs is relatively slim. The single statement referring to such alternatives on l. 680 ("While these approaches are more physiologically accurate and their predictions compare well with
observed trends, they do not guarantee that the water use is optimal over the whole optimization period.") does not do it justice in my view. I want to avoid a more fundamental debate over the "constrained water" assumption of Cowan &*

*Farquhar (1977) (How can a plant know in advance how long the current dry-down will last? How can it know how to optimally make use of available water from now until the [future] end of the dry-down? Why wouldn't it be advantageous for a competitor to consume water immediately rather than save it for the future?), but the justification of not discussing*
*alternative modelling approaches by stating that "they do not guarantee that the water use is optimal over the whole optimization period" seems unfair - particularly in view of the argument that I want to avoid getting into ;-).*

We have expanded Section 4.4 in the Discussion as detailed in our reply to comment (2), and elaborated on the 'philosophical' difference between the instantaneous maximization *vs.* optimal control approaches, also in Section 4.4:
"While these approaches are physiologically accurate in the way they balance instantaneous C gains and losses and their predictions compare well with observed trends, they do not guarantee that the water use is optimal over a given time interval. In other words, instantaneous maximization models rest on the assumption that future C gains are so uncertain that maximizing short-term gains is more convenient (in an evolutionary sense). In contrast, models based on optimal control theory rest on the assumption that future gains are expected because climatic conditions are to some degree predictable (rain
on average occurs every td days), or that plant responses have been adapted to 'anticipate' these long-term, probabilistic conditions. These approaches can be seen as end-member cases along a continuum of possible optimization strategies."

*(4) The present manuscript is heavy on algebra. I understand that this is central to the reasoning of the presented analysis, but I recommend that all efforts be made that this manuscript can be read and its reasoning intuitively understood without*
*deciphering the algebra. In general, reasonable efforts should be made to reduce the algebra, possibly relegating parts to the Appendix, while still maintaining the essential descriptions. Sorry that my point here is not more specific, but I recommend that the presentation of the science be presented to appeal to the widest possible audience.*

We moved most of the material from Section 2.3 "Optimal stomatal control models" to Appendix B, leaving a streamlined
text presenting the results in an intuitive way. We left Sections 2.1 and 2.2 as they are, since they provide basic equations or final analytical results (not derivations). Other minor re-arrangements were done to improve clarity and flow (e.g., the soil moisture balance is now described in the section on optimization models, as it is most relevant there).

*MINOR*

*The specific scientific question and scope of the manuscript and the model investigation is not immediately clear. The last sentence of the abstract points to the essence being the coordination of physiology and morphology in their response to eCO2. Then, the question is stated more precisely as "... but it is not clear if and under which conditions these two effects balance out." (l. 44). Is this the central question? Does the paper answer this question? If so, could an answer to that*
*question be given more clearly in the Conclusion section? The introduction shifts attention to other questions (l. 65, l. 105-*

*107) and answers to these make up much of the Conclusions section instead. This comes at the cost of a not so well-defined scope overall.*

We re-phrased the research questions as follows, including a statement of our further methodological aim to compare model predictions and discuss the limitations of the PETA and optimization models:

"1.       How do physiological (stomatal conductance) and morphological (leaf area) adjustments coordinate to determine leaf and canopy gas exchange rates under atmospheric $CO_2$ concentrations?

2.       How do these physiological and morphological adjustments vary under combined changes in $CO_2$ concentration and atmospheric or soil drought?

By comparing the predictions of the PETA and optimization models, a theoretical perspective on these questions is offered, while identifying advantages and limitations in these different modelling approaches."

In the conclusions, we more clearly answer these questions: "Both models predict low sensitivity of canopy transpiration rates to a changing climate, indicating that morphological adjustments (leaf area increase) compensate physiological adjustments (stomatal closure)." The very last sentence in the Conclusions re-states the same message: "Overall, these results imply that physiological and morphological traits acclimate to changing environmental conditions in a coordinated manner to ensure that limiting resources such as water are used efficiently."

*The different model variants could be better linked with specific hypotheses about controls and mechanisms determining stomatal responses to eCO2. Are there specific questions to be answered by comparing predictions from the different models?*

The differences between the stomatal optimization model variants are presented in Fig. 3 and commented upon in the Discussion. Because the optimization approach surrogates detailed physiological processes by assuming an ecological 'goal' function, it is difficult to link these model variants to specific mechanistic hypotheses. Therefore, identifying controlling mechanisms might be quite speculative and we would prefer to avoid that.

*l. 39 "stimulates plant growth and thus increases leaf area": Is increased leaf area a consequence of stimulated growth?*

Higher net assimilation provides resources for faster growth, and hence also higher leaf area (assuming resources are invested both above and belowground). No changes were made in response to this comment.

*l. 41 "open canopies": A dependence of the eCO2 effect on leaf area subject to initial leaf area (open canopy) is mentioned throughout the manuscript. In view of canopies being open due to water limitation, nutrient limitations, low temperatures, or*

*simply due to young age, is often not specified, but may be relevant for responses and certainly for underlying mechanisms. Could references to "open canopies" be made more specific throughout?*

Clarifications to this regard have been added in Section 2.2 (also in response to a comment by Reviewer #1): "This index represents how far vegetation is from the maximum $L$ expected for that location. High $\alpha$ indicates an old stand or in general a stand with $L$ close to the maximum, where additional leaf area increases are not possible."

*l. 73-74: "The model is based ..." Add: ... and the assumption that vegetation in water-limited regions makes full use of a constrained flux of water (~precipitation).*

Only the 2013 formulation of the PETA hypothesis relies on this assumption, and we use the 2017 version, so it might be confusing to include this assumption here. No changes were made in response to this comment.

*l. 83: "Stomatal optimization" models are referred to as models relying on the "Lagrange multiplier" $\lambda$. This seems to be a too narrow definition of "stomatal optimization". In my understanding, models that predict stomatal responses to changes*

*along the soil-plant-atmosphere continuum may be considered here too (e.g., Sperry et al., 2016; Wolf et al., 2016).*

We added in the Introduction: "Among the numerous optimization-based models available (Mencuccini et al., 2019; Wang et al., 2020, and references therein), we focus here on those formulated as an optimal control problem in which stomatal conductance is solved through time." (See also response to comment (2).)

*Table 1: Confusing use of 'T' in T_a and T_d, while the two 'T' are different variables with different units. E_SR not explained.*

The symbol used for the dry period duration was changed to $t_d$; the symbol $E_{SR}$ was changed to $E_W$.

*l. 147: Spell out that ci:ca is assumed to remain constant under eCO2.*
*l. 150: "A_L is a linear function of ci but with a declining slope at high CO2 concentration". This seems to be a contradiction in itself. Either it's linear or has a varying slope.*

A clarification was added to address both comments: "As a result, $A_L$ is a linear function of $c_i$, but the slope of the relation decreases with increasing atmospheric $CO_2$ concentration; moreover, this approximation allows retaining variations in $c_i/c_a$ with $c_a$ (Katul et al., 2010)."

**References**

Bassiouni, M. and Vico, G.: Parsimony vs predictive and functional performance of three stomatal optimization principles in a big-leaf framework, New Phytol., 231, 586–600, https://doi.org/10.1111/nph.17392, 2021.

Buckley, T. N. and Schymanski, S. J.: Stomatal optimisation in relation to atmospheric $CO_2$, New Phytol., 201, 372–377, https://doi.org/10.1111/nph.12552, 2014.

Donohue, R. J., Roderick, M. L., McVicar, T. R., and Yang, Y. T.: A simple hypothesis of how leaf and canopy-level
transpiration and assimilation respond to elevated $CO_2$ reveals distinct response patterns between disturbed and undisturbed vegetation, J. Geophys. Res.-Biogeosciences, 122, 168–184, https://doi.org/10.1002/2016jg003505, 2017.

Huang, C.-W., Domec, J.-C., Palmroth, S., Pockman, W. T., Litvak, M. E., and Katul, G. G.: Transport in a coordinated soil-root-xylem-phloem leaf system, Adv. Water Resour., 119, 1–16, https://doi.org/10.1016/j.advwatres.2018.06.002, 2018.

Joshi, J., Stocker, B. D., Hofhansl, F., Zhou, S., Dieckmann, U., and Prentice, I. C.: Towards a unified theory of plant
photosynthesis and hydraulics, Rev., 2022.

Katul, G., Manzoni, S., Palmroth, S., and Oren, R.: A stomatal optimization theory to describe the effects of atmospheric $CO_2$ on leaf photosynthesis and transpiration, Ann. Bot., 105, 431–442, 2010.

Mencuccini, M., Manzoni, S., and Christoffersen, B.: Modelling water fluxes in plants: from tissues to biosphere, New Phytol., 222, 1207–1222, https://doi.org/10.1111/nph.15681, 2019.

Prentice, I. C., Dong, N., Gleason, S. M., Maire, V., and Wright, I. J.: Balancing the costs of carbon gain and water transport: testing a new theoretical framework for plant functional ecology, Ecol. Lett., 17, 82–91, https://doi.org/10.1111/ele.12211, 2014.

Sperry, J. S., Venturas, M. D., Anderegg, W. R. L., Mencuccini, M., Mackay, D. S., Wang, Y., and Love, D. M.: Predicting stomatal responses to the environment from the optimization of photosynthetic gain and hydraulic cost, Plant Cell Environ.,
40, 816–830, https://doi.org/10.1111/pce.12852, 2017.

Stocker, B., Wang, H., Smith, N., Harrison, S., Keenan, T., Sandoval, D., Davis, T., and Prentice, I.: P-model v1.0: an optimality -based light use efficiency model for simulating ecosystem gross primary production, Geosci. MODEL Dev., 13, 1545–1581, https://doi.org/10.5194/gmd-13-1545-2020, 2020.

Wang, Y., Sperry, J. S., Anderegg, W. R. L., Venturas, M. D., and Trugman, A. T.: A theoretical and empirical assessment
of stomatal optimization modeling, New Phytol., n/a, https://doi.org/10.1111/nph.16572, 2020.